# Tongue immune compartment analysis reveals spatial macrophage heterogeneity

Ekaterini Maria Lyras[1], Karin Zimmermann[1], Lisa Katharina Wagner[1], Dorothea Dörr[1,2], Christoph SN Klose[3], Cornelius Fischer[1], Steffen Jung[4], Simon Yona[5], Avi-Hai Hovav[5], Werner Stenzel[6], Steffen Dommerich[7], Thomas Conrad[1], Achim Leutz[1,2], Alexander Mildner[1,8,9]*

[1]Max-Delbrück-Center for Molecular Medicine Berlin, Berlin, Germany; [2]Institute of Biology, Humboldt University of Berlin, Berlin, Germany; [3]Universitätsmedizin Berlin, corporate member of Freie Universität Berlin and Humboldt-Universität zu Berlin, Department of Microbiology, Infectious Diseases and Immunology, Charité Berlin, Berlin, Germany; [4]Weizmann Institute of Science, Rehovot, Israel; [5]Institute of Dental Sciences, The Hebrew University of Jerusalem, Jerusalem, Israel; [6]Universitätsmedizin Berlin, corporate member of Freie Universität Berlin and Humboldt-Universität zu Berlin, Department of Neuropathology, Charité Berlin, Berlin, Germany; [7]Universitätsmedizin Berlin, corporate member of Freie Universität Berlin and Humboldt-Universität zu Berlin, Department of Otorhinolaryngology, Charité Berlin, Berlin, Germany; [8]InFLAMES Research Flagship Center, University of Turku, Turku, Finland; [9]Institute of Biomedicine, Medicity, University of Turku, Turku, Finland

*For correspondence:
alexander.mildner@utu.fi

**Abstract** The tongue is a unique muscular organ situated in the oral cavity where it is involved in taste sensation, mastication, and articulation. As a barrier organ, which is constantly exposed to environmental pathogens, the tongue is expected to host an immune cell network ensuring local immune defence. However, the composition and the transcriptional landscape of the tongue immune system are currently not completely defined. Here, we characterised the tissue-resident immune compartment of the murine tongue during development, health and disease, combining single-cell RNA-sequencing with in situ immunophenotyping. We identified distinct local immune cell populations and described two specific subsets of tongue-resident macrophages occupying discrete anatomical niches. *Cx3cr1*+ macrophages were located specifically in the highly innervated lamina propria beneath the tongue epidermis and at times in close proximity to fungiform papillae. *Folr2*+ macrophages were detected in deeper muscular tissue. In silico analysis indicated that the two macrophage subsets originate from a common proliferative precursor during early postnatal development and responded differently to systemic LPS in vivo. Our description of the under-investigated tongue immune system sets a starting point to facilitate research on tongue immune-physiology and pathology including cancer and taste disorders.

## Editor's evaluation

Here, the authors map the cellular landscape in the tongue, an understudied immunological organ, with a main focus on tissue-resident myeloid cells under homeostatic and inflammatory conditions. They identify two major subsets of macrophages, which occupy distinct anatomical niches and develop from local precursors, while under immune compromised conditions they can also be replenished from circulating hematopoietic precursors. These findings provide an important basis for future investigations of the tongue immune function in the context of infection, inflammation, and neoplastic diseases.

## Introduction

The tongue is a highly innervated muscular organ with functions in articulation, mastication, and taste perception. Located at the entrance of the gastrointestinal tract, the tongue is constantly exposed to dietary and airborne antigens and therefore acts as a first-line immune organ (*Wu, 2021*). Moreover, taste sensation plays a critical role in avoidance of spoiled food and beverages. Accordingly, a tongue-resident immune network would be expected with roles in immune defense, tissue remodeling and tongue homeostasis. However, in the immunological context, the tongue is an understudied organ and the composition of tongue immune cells and their transcriptional status is largely unknown. Here, we define the immune cell landscape of the tongue with a specific focus on mononuclear phagocytes, for example, tissue-resident macrophages (TRM).

Until now, the characterization of the mononuclear phagocyte compartment of the tongue mainly focused on Langerhans cells that were first described in the mouse epithelium 40 years ago (*Burkhardt et al., 1979*) and are identified in humans by their exclusive CD1a immunoreactivity (*Darling et al., 2017*; *Cruchley et al., 1989*). The characterization of sub-epithelial macrophage subsets is much more enigmatic and so far depended on the histological examination of a few membrane markers (*Agarbati et al., 2021*; *Shigeoka et al., 2021*). For example, human CD163+ macrophages can be found in subepithelial areas of the tongue (*Shigeoka et al., 2021*), a localization that they share with CD11c+ 'dendritic' cells (*Feng et al., 2009*). Relying on single markers such as CD11c to identify cells of the 'dendritic' cell lineage is problematic since various additional cell types, including mono-cytes, macrophages, and lymphocytes, can express CD11c (*Merad et al., 2013*). Besides histolog-ical examinations of mononuclear phagocytes in the healthy tongue, the macrophage involvement in various pathological settings has also been studied. Tongue Langerhans cells were for instance shown to be critically involved in IL17-dependent antifungal immunity in the oral mucosa (*Sparber et al., 2018*), play an important role in T cell priming during squamous cell carcinoma development (*Saba et al., 2022*) and are depleted in patients with advanced-stage acquired immune deficiency syndrome (*Gondak et al., 2012*). Furthermore, an increase of activated ED1+ tongue macrophages was observed in systemic inflammation in rats (*Cavallin and McCluskey, 2005*). The recent COVID-19 pandemic has also suggested a potential link of viral infections with tongue immunity, with loss of taste being one of the hallmark symptoms (*Agyeman et al., 2020*). However, the lack of knowledge of the tongue immune cell compartment in physiology, hampers our understanding of tongue immune responses following pathogen challenge. Therefore, an unbiased characterization of the tongue immune cells is critical to classify and evaluate tissue-resident cell subsets, for example, macrophage dynamics during tongue development and pathologies.

To this end, we profiled the tongue-resident CD45+ hematopoietic cell compartment by single cell RNA-sequencing (scRNA-seq). Amongst tongue innate lymphoid cells (ILC), for example, ILC2, and the specific presence of mast cells in early postnatal tongues, we further identified two main *Irf8*-independent macrophage populations, which were characterized by *Cx3cr1* and *Folr2* expression, respectively. *Cx3cr1*-expressing macrophages were specifically enriched in the lamina propria of the tongue and were detected in fungiform papillae, which harbor taste buds, but were absent from the epidermis. *Folr2*-expressing tongue macrophages localized in muscular tissue and in the lamina propria. These anatomical niches were colonized during embryonic and early postnatal development from a *Cx3cr1*-expressing precursor of high proliferation capacity. Both macrophage populations showed a robust inflammatory response after in vivo lipopolysaccharide (LPS) administration, including shared and unique pathways.

In summary, our data provide a detailed atlas of the immune cells of the tongue that will facilitate future research of this under-investigated barrier organ.

## Results

### Characterization of murine tongue hematopoietic cells

To examine the tissue-resident immune compartment of the tongue in an unbiased manner, we performed scRNA-seq of FACS-purified CD45+ hematopoietic tongue cells isolated from PBS-perfused adult wild-type C57BL/6 mice. Two biologically and technically independent 10X Chromium experiments were performed that yielded highly reproducible results (*Figure 1—figure supplement 1a+b*).

We sequenced a total of 6773 cells that clustered into 20 transcriptionally distinct subsets (*Figure 1a*) and used singleR (*Aran et al., 2019*) for cell lineage recognition (*Figure 1—figure supplement 1c*). A full list of marker genes and average expression values per cluster can be found in *Figure 1—source data 1+2*. We detected B cells (cluster 12; characterized by *Cd79a* and *Cd19*), mast cells (clusters 11 and 13; defined by *Kit* and *Ms4a2*), neutrophils (cluster 8; defined by *S100a9* and *Retnlg*), a few endothelial cells (cluster 16, which is mainly present in subsequent scRNA-seq analysis of Figure 3; defined by *Aqp1*, *Col4a1* and *Pecam1*), fibroblasts (cluster 19; defined by *Dcn*, *Peg3*, *Cald1*), and some cells that could not attributed to a specific cell subset (clusters 17). We also detected tongue-resident lymphocytes (clusters 4, 9, 14), which could be further separated into various subsets including different innate lymphoid cell (ILC) populations (ILC1 (*Cd160*, *Itga1*, *Klrk1*), nILC2 (*Arg1*, *Il1rl1*, *Gata3*) and iILC2/ILC3 (*Cysltr1*, *Gata3*, *Kit*)), γδ T cells (*Tcrg-C1*, *Trdc*, *Cd3g*), proliferating T cells (cluster 1; *Top2a*, *Tuba1b*), tissue resident αβ T cells (clusters 2 and 3; *Trac*, *Lat*, *Trbc2*), regulatory T cells (cluster 4; *Foxp3*, *Ctla4*, *Tnfrsf4*), central αβ T cells (cluster 6; *Lef1*, *Sell*, *Ccr7*) and NK cells (cluster 7; *Prf1*, *Ncr1*, *Eomes*) (*Figure 1b*). Also a small cluster of plasmacytoid dendritic cells (pDC) could be identified (cluster 6; *Ifi203*, *Ifi209*, *Bst2*). However, the majority of cells (65%; *Figure 1a*) fell into the broad category of mononuclear phagocytes and included Langerhans cells (defined by *Epcam* and *Cd9* expression; cluster 10), different classical dendritic cell (cDC) subsets (cluster 3, 15, 18 and 6), monocytes (characterized by *Ly6c2*, *Cebpb*, *Nr4a1* expression; cluster 7), proliferating myeloid precursor cells (cluster 2) and three subsets of *Cd68*-expressing macrophages (clusters 0, 1 and 5) (*Figure 1a*).

As mononuclear phagocytes would be the first responders in tongue immunity, we focused our subset analysis on this major cell compartment. First, we separated mononuclear phagocytes into macrophages and dendritic cells (DC). For this, we performed a gene set variation analysis (GSVA) (*Hänzelmann et al., 2013*) by taking advantage of published cDC and macrophage gene signatures (*Gautier et al., 2012*; *Schlitzer et al., 2015*; *Figure 1—source data 3*). The transcriptional signature of clusters 3, 15, and 18 correlated with the cDC gene signature (*Figure 1c*) and they could further be separated into cDC1 (cluster 3; defined by *Cd209a*, *Cd24a* and *Irf8*) and DC precursors (clusters 15 and 18; characterized by *Hmgb2*, *Asf1b*, *Atad2*). While singleR annotated the *Fn1*, *Lpl*, *Ear2* expressing cells with MHCII-related gene expression in cluster 6 as CD11b[+] DCs (*Figure 1—figure supplement 1c*), these cells had neither a strong DC nor a strong macrophage signature (*Figure 1c*). As the specific cDC2 gene signature (*Schlitzer et al., 2015*) was also absent in this cluster (*Figure 1—figure supplement 1d*), the precise ontogeny of these cells will need further investigation.

The GSVA analysis revealed three clusters of cells with macrophage identity, of which clusters 0 and 5 are likely end-stage differentiated macrophage subsets (*Figure 1c*). Cells in the third macrophage cluster (cluster 1) seem to be macrophages of intermediate differentiation, as their transcriptomic signature shares features with both cluster 0 and 5 (*Figure 1d+e*). Of the two terminally differentiated macrophage clusters, cluster 0 (hereafter referred to as tFOLR2-MF) expressed high levels of *Folr2*, *Lyve1*, *Pf4* and *Timd4*, while cluster 5 (hereafter referred to as tCX3CR1-MF) expressed high levels of *Cx3cr1*, *Hexb*, *Ms4a7*, *Itgax,* and *Pmepa1* (*Figure 1d+e*). A pairwise comparison of tFOLR2-MF and tCX3CR1-MF transcriptomes revealed 602 differentially expressed genes (DEGs), indicating major differences between the two tongue macrophage populations (abs(FC)>1.5and adjusted p-value <0.05) (*Figure 1f*). Importantly, tCX3CR1-MF were transcriptionally distinct from tongue Langerhans cells (tLCs; *Figure 1f*), which rather showed a cDC signature (*Figure 1c*).

We performed gene ontology (GO) enrichment analysis for the different mononuclear phagocyte clusters to gather information on possible distinct functions of these transcriptionally defined cell subtypes. Marker genes of intermediate clusters 1, 3, and 6 were not particularly enriched for any GO biological processes, which potentially reflects their intermediate gene expression signature (*Figure 1f*; the full list of GO annotation is listed in *Figure 1—source data 4*). tFOLR2-MF on the other hand were enriched for gene sets associated with blood vessel biology ('regulation of sprouting angiogenesis') and macrophage function ('cytosolic transport', 'macrophage differentiation', and 'positive regulation of nitric oxide biosynthesis'); tCX3CR1-MF showed gene enrichment for broad immune biological processes such as 'positive regulation of ERK1/2 cascade' or 'TLR3 signalling pathway' and exhibited similarities to CNS-resident microglia with gene enrichment in the biological process 'microglial cell proliferation' (*Figure 1g*).

Recent studies have indicated potential interactions of *Cx3cr1*-expressing macrophages with neurons (*Kolter et al., 2019*; *Chakarov et al., 2019*; *Wolf et al., 2017*). However, unlike these CX3CR1[+]

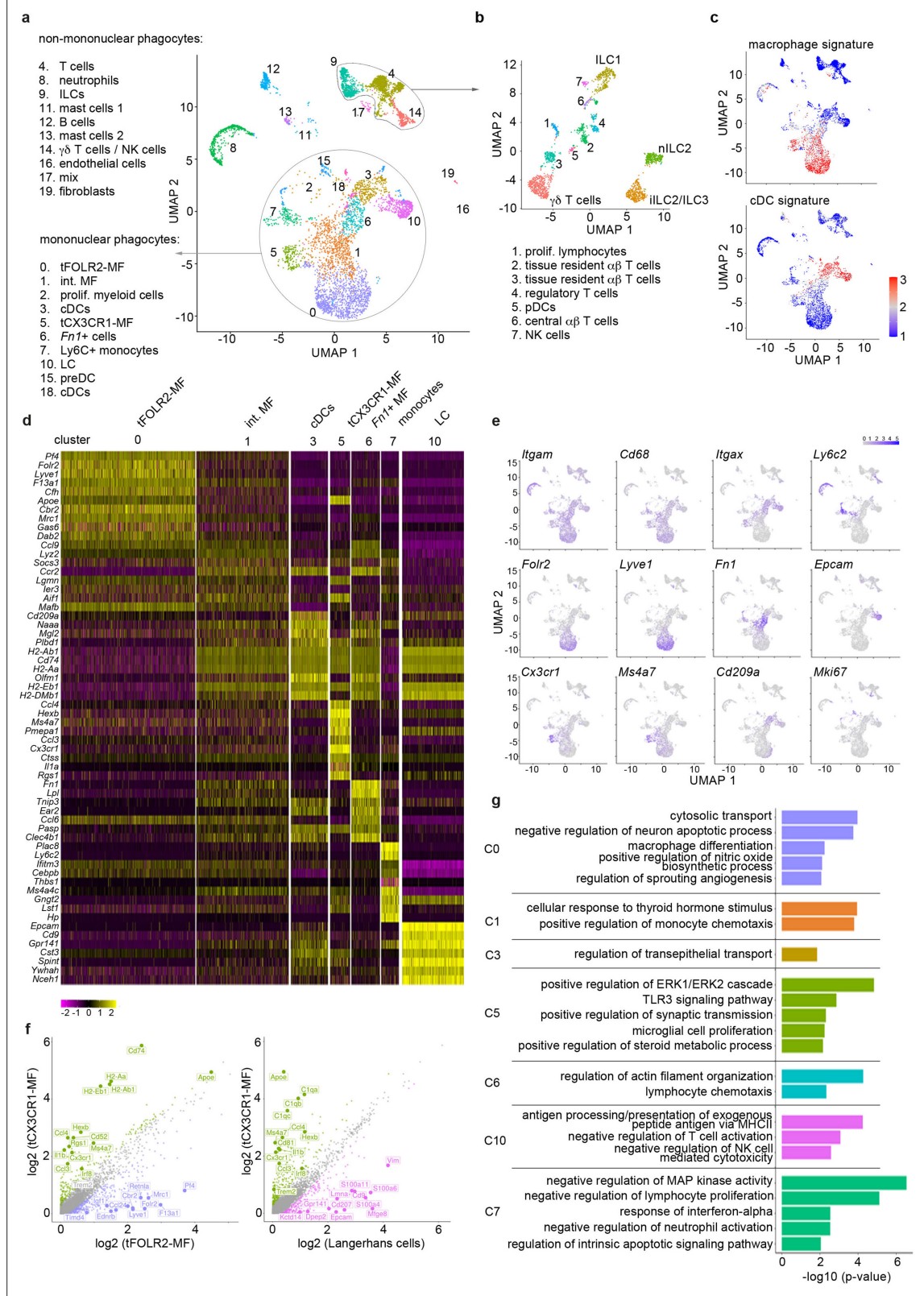

**Figure 1.** Single-cell sequencing characterization of leukocytes in the mouse tongue. (**a**) UMAP representation of 6773 sequenced tongue leukocytes from adult, female Bl6 mice (pool of n=8 mice). Data from a biologically and technically independent experiment is shown in *Figure 1—figure supplement 1a+b*. Cluster annotation was performed with SingleR (*Figure 1—figure supplement 1c*). See *Figure 1—source data 1+2* for complete gene lists and marker genes for all clusters. Abbreviations: ILCs (innate lymphoid cells); NK cells (natural killer cells); int. MF (intermediate macrophages);

*Figure 1 continued on next page*

*Figure 1 continued*

tFOLR2-MFs (tongue Folr2 +macrophages); cDCs (classical dendritic cells); tCX3CR1-MF (tongue *Cx3cr1*+macrophages); *Fn1*+ cells (Fn1-expressing mononuclear phagocytes); LC (Langerhans cells); preDC (pre-dendritic cells). (**b**) Separate UMAP representation of cells within cluster 4, 9, and 14. Additional abbreviations: pDC (plasmacytoid dendritic cells); nILC (natural ILC); iILC (induced ILC). (**c**) Gene Set Variation Analysis (GSVA) analysis was used for the discrimination of macrophages and dendritic cells. One signature gene list for macrophages (derived from *Gautier et al., 2012*) and one for cDC (derived from *Schlitzer et al., 2015*) were used to evaluate the enrichment score of each list in the 20 identified clusters. See also *Figure 1—figure supplement 1d* for cDC1 and cDC2 gene signatures and *Figure 1—source data 3* for full gene lists. Cells with the highest similarity to each respective signature are labeled red. (**d**) Heatmap of top marker genes for the main mononuclear phagocyte clusters. See *Figure 1—figure supplement 1b* for a heatmap of marker genes for all clusters. (**e**) Expression pattern of example genes laid over the UMAP from **a** for dimension reduction. (**f**) Differentially expressed genes in tCX3CR1-MF vs. tFOLR2-MF (left) and tCX3CR1-MF vs. tongue Langerhans cells (right). Indicated genes show an increased expression of >1.5 with an adjusted p-value <0.05. (**g**) Gene ontology analysis of the differentially expressed genes of the main mononuclear phagocyte clusters. Only GO annotations involved in biological processes are shown and redundant pathways were excluded from this representation. See *Figure 1—source data 4* for full list of GO terms per cluster.

The online version of this article includes the following source data and figure supplement(s) for figure 1:

**Source data 1.** Marker genes for the 19 identified clusters.

**Source data 2.** Average expression matrix of scRNA-seq data.

**Source data 3.** Core gene signatures.

**Source data 4.** GO enrichment of myeloid clusters.

**Figure supplement 1.** Characterization of adult tongue leukocytes.

brown adipose tissue and skin nerve-associated macrophages (*Kolter et al., 2019*; *Chakarov et al., 2019*; *Wolf et al., 2017*), we did not detect a specific and significant enrichment for genes involved in axon guidance, such as *Plxn4* in tCX3CR1-MF (*Figure 1—figure supplement 1e*).

Altogether, our scRNA-seq data show that the tongue harbours a wide range of tissue-resident immune cells, of which the majority belong to the mononuclear phagocyte system. We identified two terminally differentiated macrophage subsets: tCX3CR1-MF that have a transcriptomic signature associated with innate immune signaling and tFOLR2-MF that seem to function in blood vessel biology and phagocytosis.

## Tongue macrophages belong to the family of interstitial macrophages

We next established a protocol for the identification and isolation of tCX3CR1-MF and tFOLR2-MF by flow cytometry in *Cx3cr1*$^{Gfp/+}$ reporter mice (*Jung et al., 2000*). After DNase/Collagenase IV/Hyaluronidase digestion of the tongue (*Figure 2—figure supplement 1a*), we were able to detect CD64$^+$ cells that could further be separated into cells expressing high levels of *Cx3cr1*-GFP (tCX3CR1-MF) and cells that stained positive for FOLR2 (tFOLR2-MF; *Figure 2a*). tCX3CR1-MF also expressed the surface receptors CX3CR1, MHCII and CD11c, while tFOLR2-MF were additionally characterized by LYVE1 and TIMD4 expression (*Figure 2b*). These surface characteristics could also be used to identify the tCX3CR1-MF and tFOLR2-MF in WT Bl6 animals. Of note, a different digestion protocol was necessary to isolate Epcam$^+$ tLCs from the epithelial layer of the tongue (see Materials and method section; *Figure 2—figure supplement 1b*).

To identify tongue-specific signatures of tCX3CR1-MF and tFOLR2-MF and to place them in the context of macrophage biology, we compared the transcriptional profiles of FACS-purified tongue macrophages with macrophages isolated from other tissues. We FACS-isolated embryo-derived macrophages like microglia from the brain, alveolar macrophages from the lung (lung AM), F4/80$^+$ splenic red pulp macrophages (spleen RPM) and skin as well as tongue Langerhans cells (sLC and tLC, respectively). We additionally isolated heart MHCII$^+$ and MHCII$^-$ *Cx3cr1*-GFP$^+$ macrophages and MHCII$^+$ intestinal macrophages (colon MF), which are likely of monocyte origin (*Bain et al., 2014*; *Dick et al., 2018*; *Varol et al., 2007*; see *Figure 2—figure supplement 2* for gating strategy and *Figure 2—source data 1* for full read count table).

To validate the population purity, we compared expression of common macrophage-related genes in the different macrophage populations (*Figure 2c*). All cells except Langerhans cells expressed the macrophage genes *Cd68* and *Cd64*. Furthermore, *Csf1r* expression was detected in almost all macrophage subsets with particularly high levels in microglia, but not in alveolar macrophages (*Gautier et al., 2012*). *Cx3cr1*, *Lyve1* or MHCII-related genes such as *Cd74* were also expressed according to the expected pattern (*Figure 2c*), which confirmed the accuracy of our gating and sorting strategy.

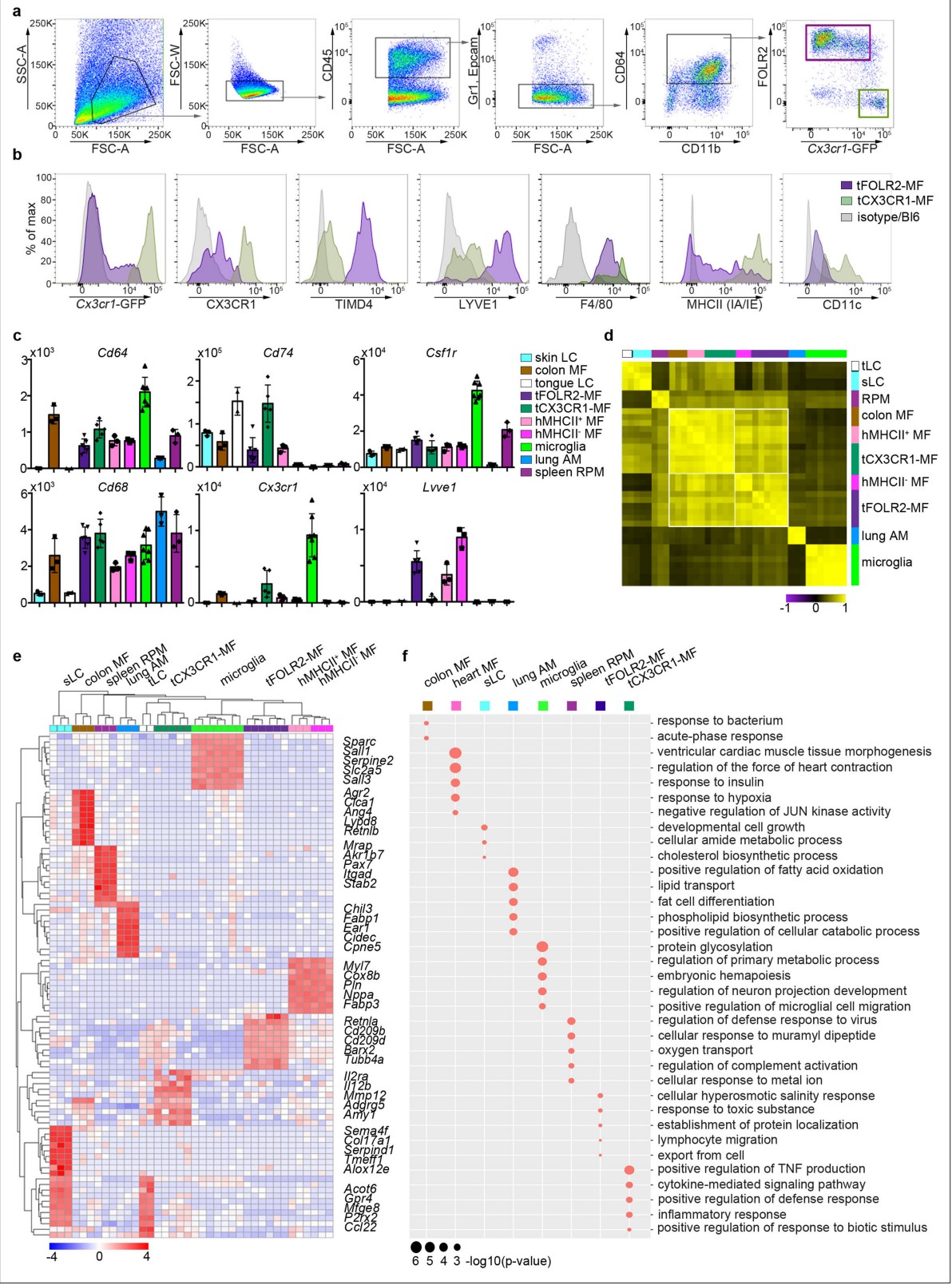

**Figure 2.** Tissue-specific transcriptomic identity of tongue macrophages. (**a**) Exemplary flow cytometry analysis and gating for tCX3CR1-MF and tFOLR2-MF in *Cx3cr1*$^{Gfp/+}$ mice. See *Figure 2—figure supplement 1* for isolation tissue procedure. (**b**) Shown are histogram expression patterns for *Cx3cr1*-GFP, CX3CR1, TIMD4, LYVE1, F4/80, MHCII and CD11c on tFOLR2-MF (violet) and tCX3CR1-MF (green). Either isotype controls or wild-type Bl6 mice were used to control for antibody stain or GFP signal, respectively. (**c**) Tissue resident macrophage populations were isolated from different tissues

*Figure 2 continued on next page*

*Figure 2 continued*

by FACS (see *Figure 2—figure supplement 2* for gating strategy; n=2-7) and analyzed by bulk RNA sequencing. Normalized read counts of important macrophage genes are shown across subsets. See also *Figure 2—source data 1* for normalized read counts. (**d**) Sample-wise expression correlation analysis of the different macrophage subsets is shown. Color code as indicated in c. (**e**) Heatmap of upregulated genes across macrophage populations. The top 10 upregulated genes per population compared to all other populations are depicted. See also *Figure 2—source data 2* for full list of upregulated genes. (**f**) Shown are GO annotations of biological processes that are enriched in specific macrophage subsets. Note that tLC showed no specific enrichment and are therefore not represented in the graph. Furthermore, MHCII- and MHCII+ heart macrophages were not separated for this analysis. Abbreviations: LC (Langerhans cells), MF (macrophages), RPM (red pulp macrophages), AM (alveolar macrophages).

The online version of this article includes the following source data and figure supplement(s) for figure 2:

**Source data 1.** Read count table bluk RNA-Seq.

**Source data 2.** GO enrichment for myeloid populations.

**Figure supplement 1.** Different isolation methods for the identification of tongue leukocytes.

**Figure supplement 2.** Gating strategy for the isolation of tissue resident macrophages.

Correlation analysis of all macrophage populations revealed that, in line with published data (*Aran et al., 2019*; *Bain et al., 2014*), classical TRM populations such as microglia, splenic macrophages, Langerhans cells and alveolar macrophages each had a very distinct expression profile, indicating the robust tissue imprinting of these cells (*Figure 2d*). Tongue macrophages on the other hand were more related to heart and intestinal macrophages, regardless of their tissue of residence. Morover, even within this group of TRM, MHCII expressing cells like heart MHCII+ macrophages, tCX3CR1-MF and colon macrophages showed a higher correlation to each other and were distinct from MHCII- cell populations, including tFOLR2-MF and heart MHCII+ macrophages (*Figure 2d*). Thus, tCX3CR1-MF and tFOLR2-MF share similarities with the two main interstitial macrophage populations that have been previously identified across various tissues (*Chakarov et al., 2019*; *Dick et al., 2022*).

We focused our analysis on up-regulated genes (FC >2; adjusted p-value <0.001) to identify a tissue-specific signature of each macrophage subset and were able to annotate previously described marker genes to macrophage populations isolated from the lung, heart, skin, brain and spleen (*Gautier et al., 2012*; *Summers et al., 2020*). tCX3CR1-MF expressed significantly higher levels of *Il2ra*, *Il12b* and *Mmp12* compared to all other tested TRM subsets, while tFOLR2-MF were characterized by the transcription of *Cd209* gene family members (*Cd209b/d/f*), *Retnla*, *Clec10a* and *Fxyd2*. The top genes for each macrophage subset are shown as a heatmap in *Figure 2e* and a full list of these up-regulated genes can be found in *Figure 2—source data 2*.

We next performed a GO enrichment analysis on these DEGs and found in agreement with previously published work (*Lavin et al., 2014*) an enrichment of GO terms that facilitate the tissue-specific function of each TRM subset (*Figure 2f*). In comparison, tCX3CR1-MF showed a strong enrichment for genes involved in inflammatory pathways, such 'positive regulation of TNF production' or 'cytokine-mediated signaling pathway', which is in line with our scRNA-seq data presented in *Figure 1*. tFOLR2-MF were characterized by weak but significant enrichment for 'cellular hyperosmotic salinity response' and 'response to toxic substances'.

Taken together, these data demonstrate that tCX3CR1-MF and tFOLR2-MF tongue macrophages belong to the family of interstitial macrophages. They fall into the two broad categories of *Cx3cr1-* and *Lyve1/Folr2/Timd4*-expressing cells (*Chakarov et al., 2019*; *Dick et al., 2022*), but also show unique transcriptomic signatures that probably reflect the requirements of their local tissue niche.

## Distinct localizations of tCX3CR1- and tFOLR2-macrophages in the adult tongue

The unique transcriptomic signatures of tCX3CR1-MF and tFOLR2-MF could indicate that these populations inhabit different microanatomical niches within the tongue. To test this hypothesis, we performed immunohistochemistry on adult mouse tongues. $Cx3cr1^{Gfp/+}$ animals were perfused and fixed tongue sections were stained with antibodies against GFP and LYVE1. Of note, we have used LYVE1 and FOLR2 markers interchangeably for the identification of tFOLR2-MF. Indeed, GFP+ cells were concentrated in the lamina propria, while LYVE1 staining was evident on cells throughout the tongue tissue with the exception of the epidermis (*Figure 3a+b*). Since lymphatic vessels also stain positive for LYVE1 (*Prevo et al., 2001*), the tissue was counterstained with antibody against CD68, a

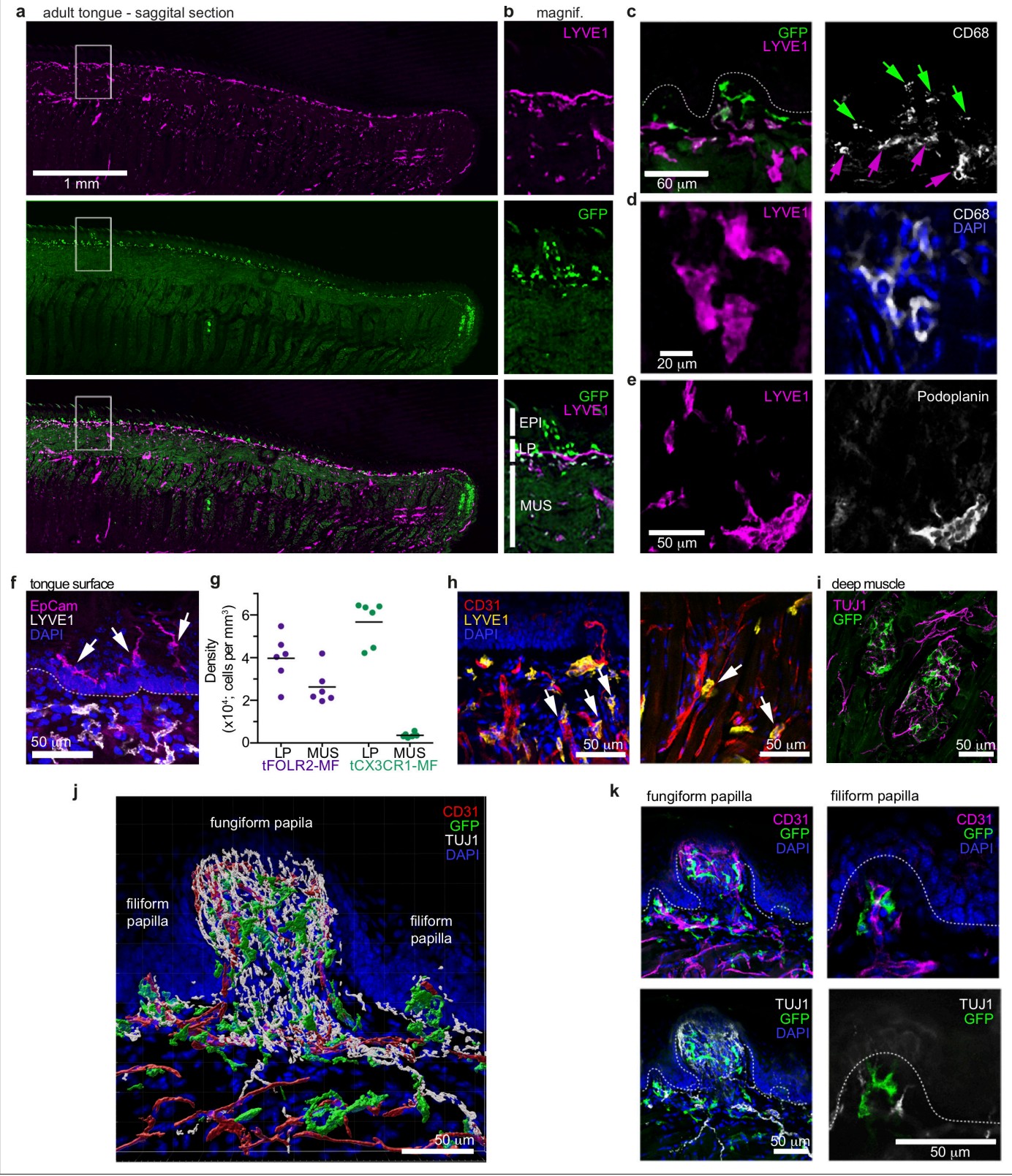

**Figure 3.** Distinct localization of mouse tongue macrophages. (**a**) Panoramic image of a tongue sagittal section from a perfused *Cx3cr1*$^{Gfp/+}$ mouse. The tongue was stained for anti-GFP (CX3CR1; green) and anti-LYVE1 (violet). (**b**) Magnification of the area depicted in (a) that includes a fungiform papilla with annotation of different tissue layers. EPI = epidermis, LP = lamina propria, MUS = muscle. (**c**) Co-staining of anti-GFP (green) and anti-LYVE1 (violet) with anti-CD68 (white). The dashed white line indicates the border between the epidermis and the lamina propria. Green arrows indicate *Cx3cr1-*

*Figure 3 continued on next page*

*Figure 3 continued*

GFP⁺CD68⁺ macrophages, while violet arrows highlight LYVE1⁺CD68⁺ macrophages. (**d**) High magnification of LYVE1⁺ cells (violet) stained with anti-CD68 (white) and DAPI (blue). (**e**) Discrimination of podoplanin⁺ (white) lymphatics from LYVE1⁺ (violet) podoplanin⁻ macrophages. (**f**) Anti-Epcam (violet) staining identifies Langerhans cells in the epidermis of the tongue. Sections were stained with anti-Lyve1 (white) and DAPI (blue). (**g**) Quantification of tFOLR2-MF and tCX3CR1-MF in lamina propria (LP) and muscle layer (MUS) in adult female mouse tongues. Each dot represents one animal. See also *Figure 3—figure supplement 1* for definition of layers. (**h**) Proximity of LYVE1⁺ (yellow) cells to CD31⁺ (red) blood vessels. (**i**) *Cx3cr1*-GFP⁺ (green) cell clusters could also be detected in innervated Tuj1⁺ (violet) areas in posterior regions of the tongue. (**j**) 3D reconstruction of *Cx3cr1^Gfp/+* tongue tissue sections stained for anti-CD31 (red), anti-Tuj1 (white), anti-GFP (*Cx3cr1*; green) and DAPI (blue) using Imaris. (**k**) Localization of tCX3CR1-MF in fungiform (left) and at the base of filiform (right) papillae. Sections were stained for anti-CD31 (violet), anti-Tuj1 (white), anti-GFP (*Cx3cr1*; green) and DAPI (blue). Shown in this figure are representative images from n=6 experiments.

The online version of this article includes the following figure supplement(s) for figure 3:

**Figure supplement 1.** Quantification of tongue macrophages.

common pan-macrophage marker. Thus, we could identify tCX3CR1-MF as double-positive CD68⁺*Cx-3cr1*-GFP⁺ cells in the lamina propria, but not in the tongue epidermis or muscle (*Figure 3c*) and tFOLR2-MF as double-positive CD68⁺LYVE1⁺ cells in the lamina propria and the underlying muscle (*Figure 3d*). Of note, tFOLR2-MF were morphologically distinct from LYVE1⁺Podoplanin⁺ lymphatics (*Figure 3e*). EPCAM⁺ Langerhans cells with a ramified morphology localized exclusively in the epidermis of the tongue (*Figure 3f*).

To better characterize the tissue localisation of tCX3CR1-MF and tFOLR2-MF, we quantified their distribution in different layers of the tongue. CD68⁺LYVE1⁺ double-positive tFOLR2-MF localized in the muscular layer as well as in the lamina propria (*Figure 3g+h* and *Figure 3—figure supplement 1*), while CD68⁺*Cx3cr1*-GFP⁺ double-positive tCX3CR1-MF were only detected in the lamina propria and were virtually absent in muscular tissue (*Figure 3g*). However, clusters of *Cx3cr1*-GFP⁺ cells could be detected in the posterior part of the tongue, along TUJ1⁺ nerves that possibly cater to circumvallate and foliate papillae (*Figure 3i*) and in innervated areas of the deep muscle, along the chorda tympani branch of the facial nerve. tCX3CR1-MF were present at the base of both filiform and fungiform papillae (which harbour taste buds) and within the lamina propria, which is densely innervated by sensory fibres (*Figure 3j+k*).

Thus, we show here that tongue tCX3CR1-MF and tFOLR2-MF inhabit distinct anatomical regions of the tongue. tCX3CR1-MF localized in the highly innervated lamina propria at the base of filiform papillae and within fungiform papillae, while tFOLR2-MF can additionally be found in deeper layers, often in proximity to blood vessels.

## Response of tongue macrophages to systemic inflammation

Regardless of their tissue-specific roles in homeostasis, macrophages usually also function as first-line responders to pathogens, especially in barrier organs. The observed differences in the homeostatic inflammatory signature between tCX3CR1-MF and tFOLR2-MF (*Figure 2f*) led us to hypothesize that these cells might respond differently in inflammation. It has already been shown that a systemic inflammatory stimulus, such as the intraperitoneal (i.p.) injection of bacterial endotoxin lipopolysaccharide (LPS), can lead to increased expression of TNFa in type II taste cells and thereby attenuates taste bud cell homeostasis (*Feng et al., 2012*; *Cohn et al., 2010*). We therefore tested the inflammatory response of tongue immune cells to systemic LPS. Mice were challenged i.p. with LPS and, 6 hr after injection, we observed reduction of LYVE1 surface expression on tongue CD11b⁺ CD64⁺ immune cells (*Figure 4a*), which reflects the susceptiblity of LYVE1 to inflammatory shedding and degradation (*Johnson et al., 2007*; *Kim et al., 2021*). We proceeded to investigate potential cell responses at a higher resolution with scRNA-Seq. In total, we sequenced 8165 hematopoietic cells from a pool of 6 mice 6 hr after LPS (i.p.). Integration of the LPS data with the data from steady state mice indicated that all cell populations were present 6 hr after LPS injection (*Figure 4b*). In general, LPS injection led to an increase of inflammatory gene expression such as *Oasl1* and *Ifi204* across all cells of the tongue immune system (*Figure 4c*), which indicates a widespread response to systemic LPS as suggested previously (*Feng et al., 2012*; *Cohn et al., 2010*).

We focused on tongue-resident macrophages of cluster 0 (tFOLR2-MF) and cluster 5 (tCX3CR1-MF) and examined DEGs between the steady state and LPS conditions in these two populations. 211 DEGs were identified in tFOLR2-MF (the full list of DEGs can be found in *Figure 4—source data 1*), of

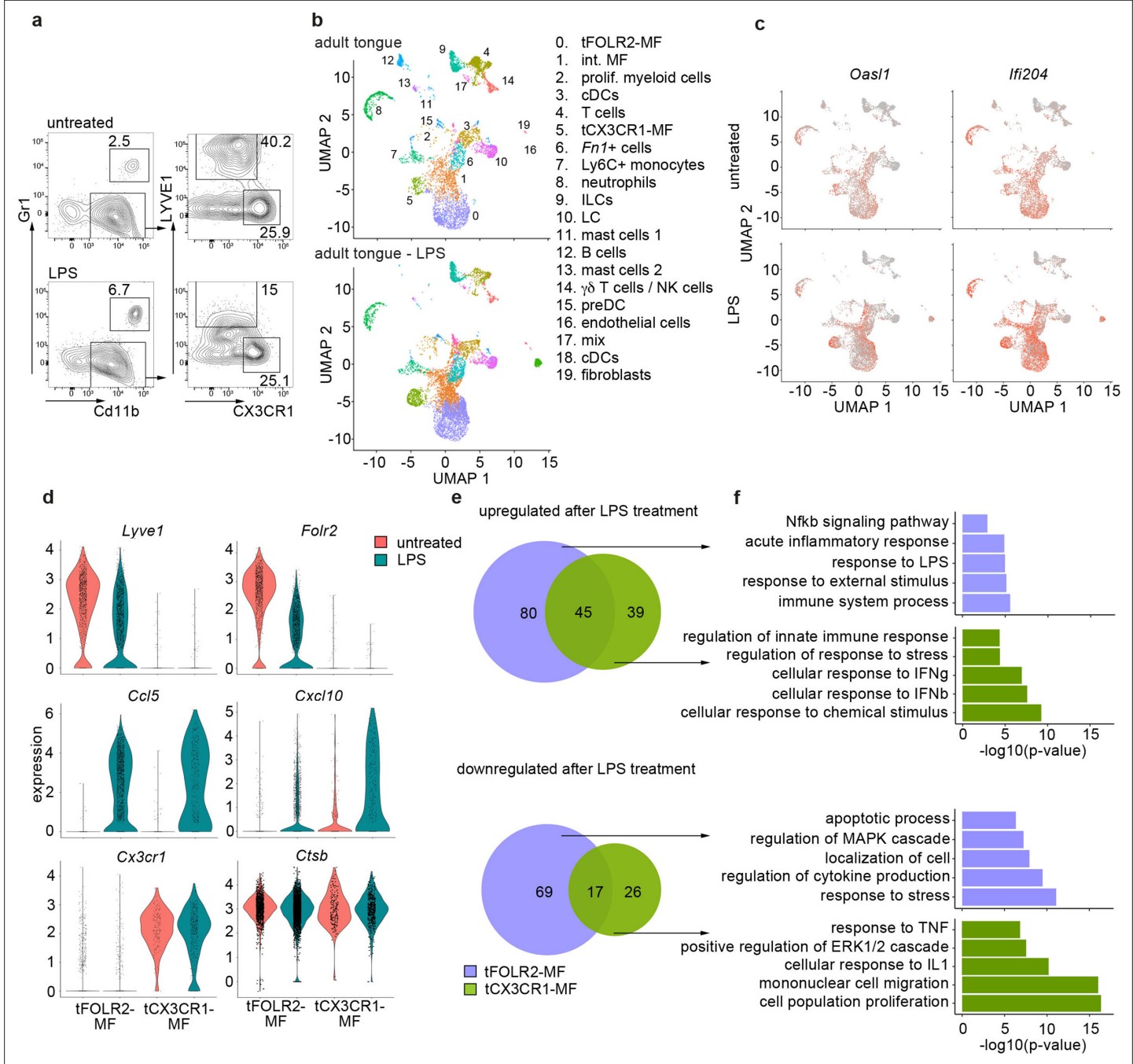

**Figure 4.** Inflammatory response of tongue macrophages to systemic LPS challenge. (**a**) Female Bl6 mice were intraperitonally injected with 1 mg/kg LPS and analyzed 6 hr after injection by flow cytometry. Note the absence of LYVE1 surface expression in CD11b⁺ CD64⁺ tongue macrophages after LPS treatment. The experiment was performed twice with n=3–6 animals. (**b**) Tongue CD45⁺ leukocytes were FACS purified 6 hours after injection of LPS and subjected to scRNA-seq (pool of 6 mice). In total, 8165 LPS-exposed tongue leukocytes were sequenced and the data was integrated into the untreated adult analysis shown in *Figure 1* for population comparison. Shown are UMAPs for untreated adult cells (top) and LPS-treated cells (bottom). (**c**) Gene expression pattern of *Oasl1* and *Ifi204* in untreated and LPS-treated tongue leukocytes. (**d**) Violin blots showing marker gene expression patterns in untreated (red) and LPS treated (green) tFOLR2-MF and tCX3CR1-MF. (**e**) Number of up- and down-regulated genes in tFOLR2-MF (violet) and tCX3CR1-MF (green) after LPS injection. The full list of DEGs can be found in *Figure 4—source data 1*. (**f**) GO enrichment analysis of the DEGs. Only GO annotations involved in biological processes are shown.

The online version of this article includes the following source data for figure 4:

**Source data 1.** List of differential expressed genes between untreated and LPS treated myeloid cells.

which 125 genes were upregulated (e.g. *Il1b*, *Relb* and *Slfn4*) and 86 genes downregulated after LPS injection (e.g. *Folr2*, *Lyve1*, and *Klf4*; *Figure 4d+e*). Interestingly, many of the tFOLR2-MF signature genes (i.e. *Folr2* and *Lyve1*) were downregulated after LPS exposure (*Figure 4c*). In tCX3CR1-MF we detected 127 DEGs between the physiological and the pathological state of which 84 were upregulated (e.g. *Ifit2*, *Lgals3*, and *Usp18*) and 43 were downregulated (e.g. *Lyz2*, *Ccr2*, and *Cd9*). LPS induced upregulation of 45 common genes (e.g. *Cxcl10*, *Ccl5*, *Il1rn*, and *Gbp2*) and downregulation of 17 common genes in both tongue macrophage subsets (e.g. *Fcrls*, *S100a10*, *Lyz1*, and *Retnla*; *Figure 4d+e*).

GO enrichment analysis was used to explore potential signaling differences of tFOLR2-MF and tCX3CR1-MF in response to systemic LPS. Shared upregulated genes in the two subsets were involved in 'defense response', 'response to cytokine', and 'response to bacterium'. Genes that were only upregulated in tCX3CR1-MF macrophages were particularly enriched for GO terms associated with type I interferon signaling (*Figure 4f*). On the other hand, tCX3CR1-MF showed downregulation of genes involved in 'mononuclear cell migration' and 'cell population proliferation' after LPS exposure (*Figure 4f*). tFOLR2-MF showed the specific upregulation of genes involved in ,immune system process' and ,Nfkb signaling', while they downregulated in response to LPS the 'response to stress', ,localisation of cell' apoptotic processes.

Thus, both tFOLR2-MF and tCX3CR1-MF were activated by systemic administration of bacterial components (LPS). However, they responded differently, with strong type I interferon signaling response characterizing tCX$_3$CR1-MF and a more 'classical' Nfkb-mediated macrophage response seen in tFOLR2-MF.

## Distribution and subset analysis of tongue macrophages during development

We next investigated the spatiotemporal distribution of tongue macrophages over development in an effort to shed light on the origin and timeline of establishment of tFOLR2-MF and tCX3CR1-MF populations. First, we isolated leukocytes from *Cx3cr1*[Gfp/+] reporter mice at different ages and performed flow cytometry to detect CD11b+CD64+ macrophages that we could further separate according to *Cx3cr1*-GFP expression and FOLR2 immunoreactivity. We used FOLR2 as a marker since macrophages at this developmental stage do not show LYVE1 surface expression. At embryonic day 17.5 (E17.5), all tongue macrophages were characterized by high *Cx3cr1*-GFP expression (*Figure 5a*). Of these *Cx3cr1*-GFP+ cells, two-thirds additionally expressed FOLR2 (G2 in *Figure 5a*). Similar proportions of macrophage subsets were observed in the mouse tongue at postnatal day 3 (p3). At this timepoint, an intermediate *Cx3cr1*-GFP[int] FOLR2+ subset was also present (G3 in *Figure 5a+b*). In subsequent developmental stages (p11 and p28), the proportion of double-positive *Cx3cr1*-GFP+FOLR2+ cells progressively decreased and two main *Cx3cr1*-GFP+FOLR2- and *Cx3cr1*-GFP[int]FOLR2+ macrophage populations (corresponding to tCX3CR1-MF and tFOLR2-MF, respectively) were established by 8 weeks of age (*Figure 5b*).

To correlate the flow cytometry data to spatial localization, we performed immunohistochemical analysis of tongue sections from *Cx3cr1*[Gfp/+] mice during early developmental stages. As mentioned above, tongue macrophages did not express *Lyve1* during development and the FOLR2 antibody used for FACS did not work for immunohistochemical staining of the tissue. We thus relied on *Cx3cr1*-GFP as a marker of macrophages. Tissue sections were counterstained with antibodies against CD31 to identify blood vessels, and, since various nerves (e.g. VII[th], IX[th] and X[th] cranial nerve ganglia) innervate the tongue tissue during embryogenesis, we also stained tissue sections for anti-beta Tubulin III (TUJ1) to visualize neurons. At E14.5 *Cx3cr1*-GFP+ macrophages could be detected throughout the whole tongue tissue with no specific localization pattern (*Figure 5c*). This continued through postnatal day p0 and until p11, whereby *Cx3cr1*-GFP+ macrophages were still dispersed throughout the tongue but started to align along the lamina propria. At these stages, taste bud maturation is observed (*El-Sharaby et al., 2001*; *Harada et al., 2000*) and we detected the first *Cx3cr1*-GFP+ macrophages in proximity to fungiform papillae (*Figure 5c*).

These data demonstrate the progression of a dynamic tongue macrophage compartment during mouse tongue development and that the specific localization of tongue macrophages in distinct anatomical niches is only established during postnatal stages.

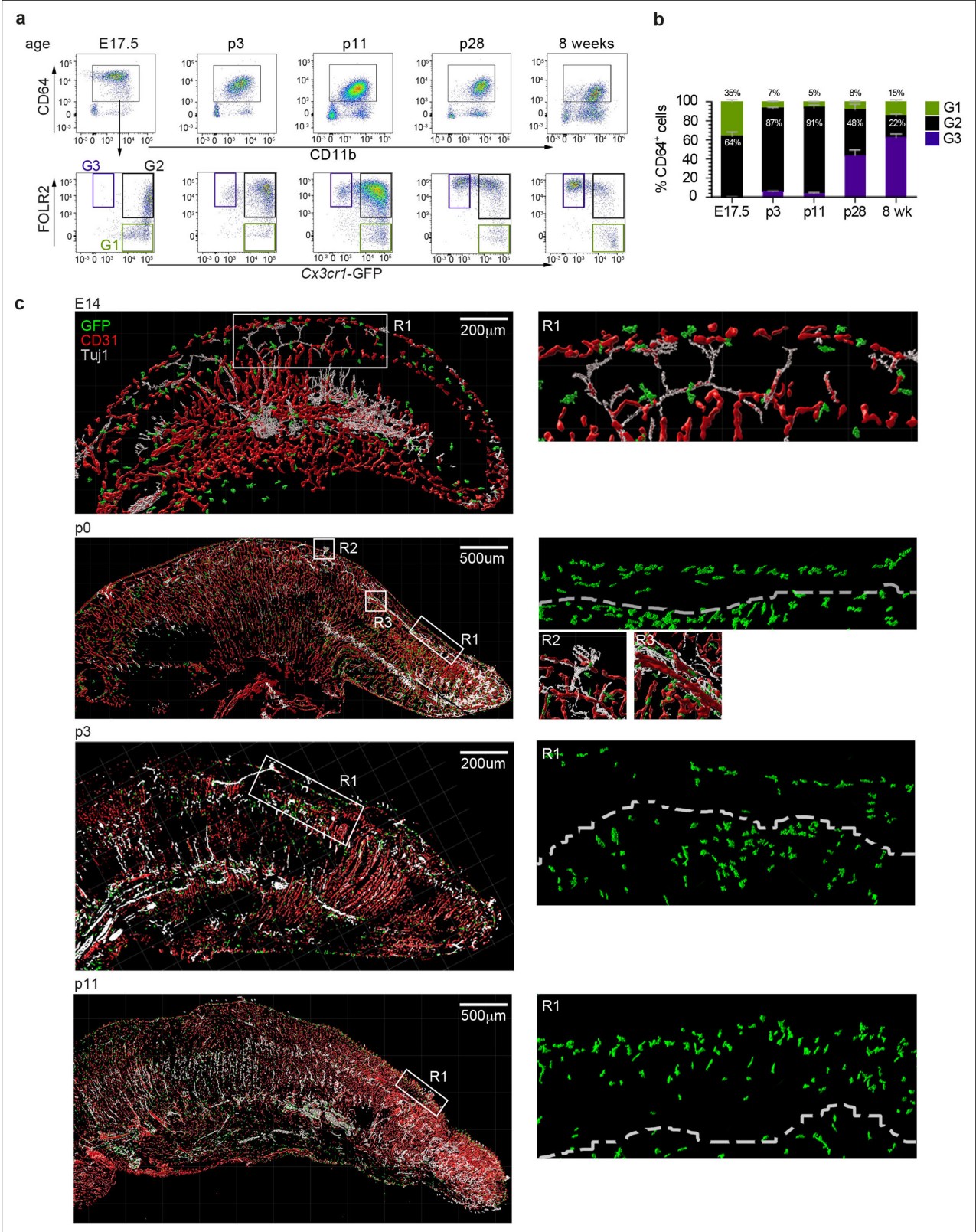

**Figure 5.** Tongue macrophage distribution during development. (**a**) Representative flow cytometry analysis of tongue leukocytes isolated from *Cx3cr1*[Gfp/+] mice at different embryonic (E) and postnatal (p) days of development and 8 week-old adults. Cells were pre-gated for CD45+ Epcam- Gr1-. (**b**) Proportions of *Cx3cr1*-GFP+ FOLR2- (G1) *Cx3cr1*-GFP+ FOLR2+ (G2) and *Cx3cr1*-GFP- FOLR2+ (G3) cells out of CD64+ cells. Five to 7 animals per time point were used. (**c**) 3D reconstructions of whole tongue sagittal sections (left) and magnifications (right) stained with anti-GFP (green), anti-CD31 (blood

*Figure 5 continued on next page*

*Figure 5 continued*

vessels; red) and anti-TUJ1 (neurons; white) from E14, p0, p3, and p11 Cx3cr1$^{Gfp/+}$ reporter mouse tongues. White dashed line in magnifications (right) represents the barrier between the epithelium layer and the muscle. The experiment was repeated at least 3 times.

## Molecular characterization of tongue macrophages during development

The histological analysis reveals the unordered distribution of *Cx3cr1*-GFP$^+$ macrophages during early developmental stages towards distinct localization at adulthood. However, it is not clear from our histological data whether this reflects a sequential developmental maturation of macrophages or rather a progressive exchange of embryonic populations with adult, bone marrow-derived monocytes.

To gain further insight into this question, we investigated the tongue immune compartment from mice at post-natal day 3 (p3) with scRNA-seq. We chose p3 since it is the stage at which we could first identify *Cx3cr1*-GFP$^{int}$FOLR2$^+$ macrophages by flow cytometry (**Figure 5b**). We sequenced 13,898 cells (from a pool of n=7 mice) and overlaid the data on the scRNA-seq data derived from adult mice (**Figure 6a**). All adult tongue immune cell populations were also present at p3, however, we noticed major differences in the frequency of several populations. Notably, p3 tongues harbored more mast cells (clusters 11 & 13) and fewer *Fn1*+ myeloid cells (cluster 6) than adult tongues (**Figure 6b**). We also detected a newly appearing cluster of cells in p3 tongues (cluster 2), which was almost absent in adult mice (**Figure 6b**). These cluster 2 cells expressed high levels of proliferation-related genes, including *Top2a*, *Ccnb2*, *Mki67*, and further showed expression of typical macrophage lineage genes such as *Cx3cr1*, *Tlr4*, *Pf4*, and *Lyve1*, which suggests they represent proliferating macrophage precursor cells (**Figure 6c**). Indeed, when the UMAP is projected in 3D, cluster 2 cells seemed to incorporate into tFOLR2-MF (**Figure 6d**; **Figure 6—source data 1**). To investigate the connection of cluster 2 precursors with tongue macrophages we used Slingshot, which models developmental trajectories in scRNA-seq data (**Street et al., 2018**). Slingshot analysis revealed a possible bifurcation of the precursor cells at the cluster 1 level at which trajectories either split into cluster 0 or cluster 5 cells (**Figure 6e**), which might indicate that both tFOLR2-MF and tCX3CR1-MF populations derive from a common precursor. To better understand the differences of tFOLR2-MF and tCX3CR1-MF cells at the p3 and adult state, we compared their expression profiles and performed GO enrichment analysis of the DEGs (**Figure 6—figure supplement 1**). Both tongue macrophage subsets showed in general an increased proliferative gene program at early postnatal stages, while functional biological processes such as 'positive regulation of angiogenesis' and 'cellular response to growth factor stimulus' increased in adult tFOLR2-MF (**Figure 6—figure supplement 1b**). tCX3CR1-MF on the other hand established a gene program that comprises 'antigen processing' and 'defense response to virus' during adulthood (**Figure 6—figure supplement 1d**), which indicates that the functional adaption to the niche likely takes place late during development. When we used latent time analysis to reconstruct the possible maturation of tongue macrophages, we detected a general down-regulation of MHC-related genes as tFOLR2-MF mature, which was accompanied by an increased expression of genes with STAT3 binding motif, while no clear maturation program could be detected in tCX3CR1-MF (**Figure 6—figure supplement 2**).

We confirmed the proliferation activity of macrophage precursor cells at p3 by EdU in vivo labelling. One day after injection, EdU incorporation could be readily observed in about 30–40% of CD64$^+$FOLR2$^+$ cells, while both adult tongue macrophage subsets showed no or weak signs of homeostatic proliferation (**Figure 6f**).

We then investigated whether peripheral cells like monocytes could adapt to the tongue macrophage niches in an immune compromised condition, for example, after whole body irradiation. We performed bone marrow (BM) chimeric experiments, in which CD45.1 BM cells were transferred into irradiated CD45.2 recipients. We analyzed the composition of donor versus host tongue macrophages 5 and 10 weeks after transfer. While tCX3CR1-MF were already replaced by monocyte-derived cells 5 weeks after transfer, the tFOLR2-MF still comprised 53% (+/-6% SD) donor cells and accordingly, showed a slower replacement (**Figure 6g**). At 10 weeks after irradiation, both tongue macrophage subsets were exclusively of hematopoietic stem cell (HSC) origin. These data indicate that both tongue macrophage niches can be repopulated by myeloid precursors during adulthood.

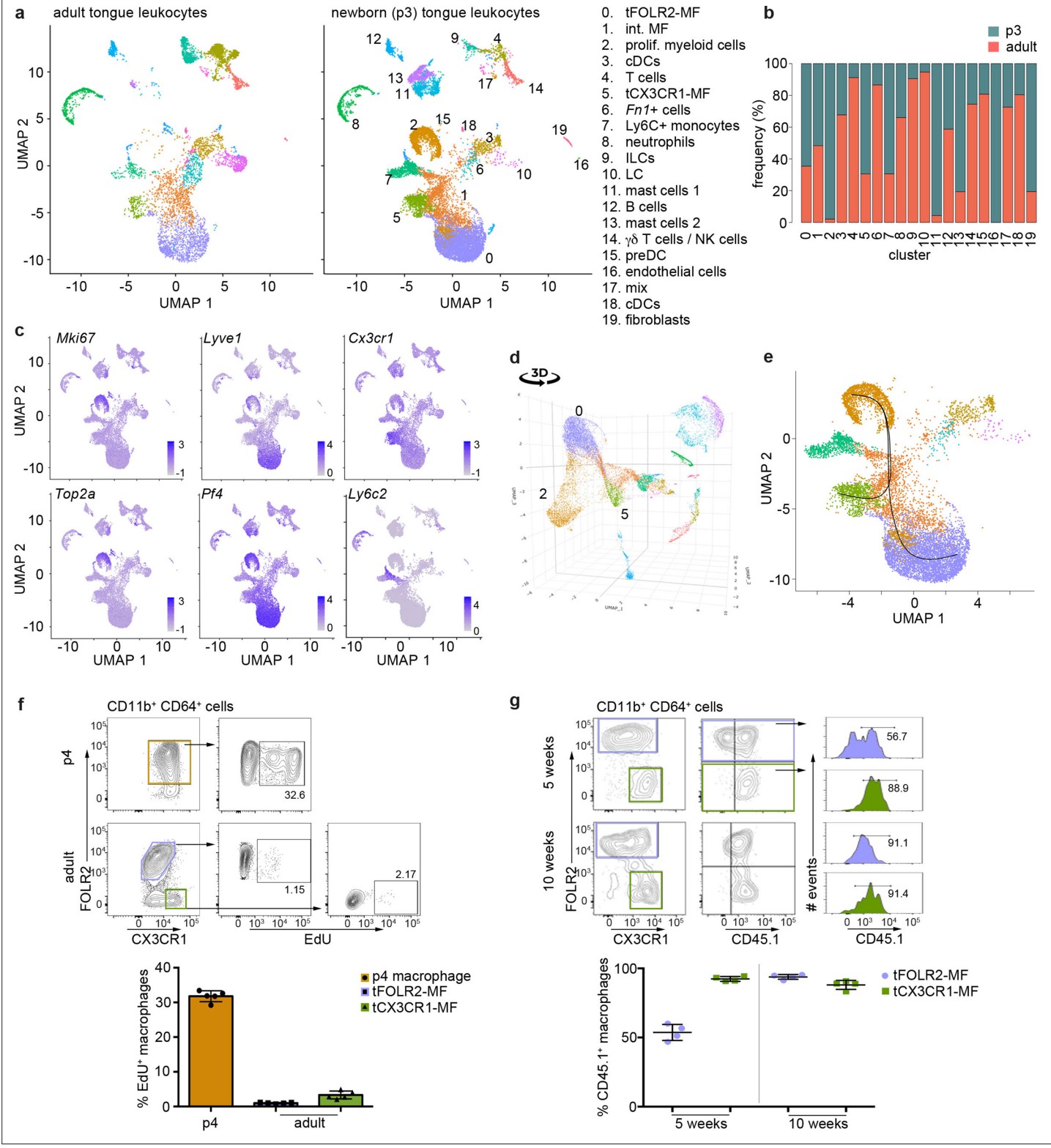

**Figure 6.** Transcriptomic analysis of hematopoietic tongue cells from newborn mice. (**a**) CD45+ tongue leukocytes were isolated by FACS from newborn p3 mice. The purified cells were subjected to scRNA-seq and a total of 13,898 cells were sequenced. The early postnatal data was integrated with the adult scRNA-seq dataset shown in *Figure 1* for population comparison. Shown are UMAPs for adult (left) and newborn cells (right). (**b**) Cell frequencies for each cell cluster in adult and newborn mice. Note that the adult analysis included enrichment for LC using a different cell isolation protocol that was not done for the p3 pup analysis (see Materials and method section) and cluster 10 LC are therefore probably underrepresented in p3 tongues.

*Figure 6 continued on next page*

*Figure 6 continued*

(**c**) Expression levels of example genes in newborn cells shown in the UMAP. (**d**) 3D UMAP visualization reveals the merging of proliferating cluster 2 cells with cluster 0 cells. The 3D html file is provided in *Figure 6—source data 1*. (**e**) Slingshot analysis was used to model the developmental trajectories of tongue macrophages. See also *Figure 6—figure supplement 1+2* for gene differences between newborn and adult tongue macrophages. (**f**) In vivo EdU proliferation assay of adult and newborn tongue leukocytes. EdU was injected and tongue cells were isolated 24 hr after injection. Representative FACS gating strategy for macrophages at each age shown above and quantification of EdU$^+$ cell fraction shown below. Each dot in the quantification represents one independent animal (n=5). This experiment was performed once. (**g**) CD45.2/2 animals were lethally irradiated (9.5 Gy) and reconstituted with CD45.1 expressing bone marrow cells. Five weeks and 10 weeks after transfer, tongue cells were isolated and analysed by flow cytometry for the distribution of CD45.1$^+$ cells within the macrophage subsets. Each dot represents one independent animal. This experiment was performed once.

The online version of this article includes the following source data and figure supplement(s) for figure 6:

**Source data 1.** 3D UMAP.

**Figure supplement 1.** Gene expression differences in tongue macrophages at p3 and adult stages.

**Figure supplement 2.** The maturation pattern of tFOLR2-MF and tCX3CR1-MF in adult tongues.

**Figure supplement 3.** *Irf8*-deficiency does not affect the tongue leukocyte composition.

To further explore the monocyte-dependency of tCX3CR1-MF and tFOLR2-MF under homeostatic conditions, we analyzed the tongue-resident immune compartment of *Irf8*-deficient mice, which lack cDC1, Ly6C$^+$ monocytes and show reduced numbers of Ly6C$^-$ monocytes (*Kolter et al., 2019*; *Kurotaki et al., 2013*; *Figure 6—figure supplement 3a+b*). We detected a slight decrease in the frequency of tCX3CR1-MF, but this population was clearly still present in the lamina propria of *Irf8*-deficient mice (*Figure 6—figure supplement 3b+c*). We also profiled the transcriptome of 9047 CD45$^+$ cells from adult *Irf8*-deficient mice (pool of n=5 mice) by scRNA-seq (*Figure 6—figure supplement 3d-f*). All tongue-resident hematopoietic cell populations that were present in WT animals could also be identified in *Irf8$^{-/-}$* mice including dendritic cells, tCX3CR1-MF and tFOLR2-MF. In addition to the unchanged mononuclear phagocyte subset composition in *Irf8$^{-/-}$* mice, the transcriptome of tCX3CR1-MF and tFOLR2-MF was also largely unaffected by the absence of the transcription factor IRF8 (9 DEGs for *Cx3cr1$^+$* and 9 DEGs for *Folr2$^+$* macrophages; *Figure 6—figure supplement 3d-f*).

Taken together, these experiments indicated that macrophage precursor cells proliferate locally and give rise to both tongue macrophage subsets. However, during adulthood and under immune compromised conditions, tCX3CR1-MF and tFOLR2-MF can be replaced by HSC-derived cells.

## Discussion

Despite the significance of the tongue as a site of interaction between microbes and the host, its cellular immune composition is not well investigated, especially not on the transcriptomic level. In light of the emerging role of different immune cells such as ILCs, regulatory T cells or macrophages in tissue remodeling, nerve surveillance and homeostatic tissue organization (*Kolter et al., 2019*; *Wolf et al., 2017*; *Wang et al., 2020*; *Monticelli et al., 2011*), we characterized the mouse tongue-resident immune cell compartment under physiological conditions. We confirmed the existence of tLCs, cDCs, and Ly6C$^+$ monocytes in the tongue (*Sparber et al., 2018*; *Park et al., 2017*; *Tanaka et al., 2016*) and additionally identified two new subsets of tongue-resident macrophages, one of which was characterized by *Folr2* and *Lyve1* expression, while the other subset expressed high levels of *Cx3cr1*, MHCII-related genes and *Itgax* (encoding CD11c). The phenotype of these two tongue macrophage populations is reminiscent of the two recently defined interstitial macrophage populations which are present across various organs (*Chakarov et al., 2019*; *Dick et al., 2022*). In agreement with this, our bulk RNA-seq analysis revealed greater transcriptional similarities of tongue macrophages with interstitial heart macrophages, for example, than with TRMs such as Langerhans cells or microglia.

Chakarov et al., further described an association of *Lyve1$^+$* macrophages with blood vessels and *Cx3cr1$^+$* macrophages with nerve fibers (*Chakarov et al., 2019*). In the adult tongue, tFOLR2-MF were distributed across both the lamina propria and the deep muscle of the tongue, but were absent from the epithelium. We could also observe the association of tFOLR2-MF to CD31$^+$ blood vessels. On the other hand, tCX3CR1-MF are largely restricted to the highly innervated lamina propria and along the fibers of the chorda tympani branch of the facial nerve. In addition, rare clusters of *Cx3cr1$^+$* macrophages could also be detected in innervated posterioir areas of the tongue. It is therefore quite likely that the previously characterized CD11c$^+$ 'dendritic cells' in the lamina propria of the human and

mouse tongues (*Feng et al., 2009*; *Mascarell et al., 2009*) correspond to the cells described here as tCX3CR1-MF. Our observation that tCX3CR1-MF acquire their disctinct localization post-natally and gradually might suggest that changes in the oral microbiota contribute to the development and alignment of tCX3CR1-MF in the lamina propria, which coincidences with lactation and weaning. Indeed, it was previously shown that local microbiota has a major impact on the development of mucosal LC (*Capucha et al., 2018*) and this might also be the case for tongue resident macrophage populations.

The proximity of tCX3CR1-MF to nerves raises the question of whether they perform specific nerve-associated functions. In the skin, large, peripheral nerve-associated *Cx3cr1*+ macrophages were recently described (*Kolter et al., 2019*) that were involved in the maintenance of myelin sheath integrity and axon sprouting after injury and had a transcriptional signature related to nervous system functions. This particular signature was absent from tCX3CR1-MF, although tCX3CR1-MF did express some genes that are enriched in microglia (for example, *Hexb*, *Apoe*) that also reside in close proximity to neurons. Various cranial nerve ganglia innervate the tongue tissue during embryogenesis and axons are guided by multiple chemoattractive factors to the tongue epithelium such as brain derived neurotrophic factor (BDNF; *Ma et al., 2009*). It is known that microglia-derived BDNF plays an important role in synapse formation and plasticity in the adult brain (*Parkhurst et al., 2013*). It is therefore possible that the subsequent postnatal maturation of taste receptor cells and the synaptic interconnectivity with neurons might be influenced by tCX3CR1-MF. The interesting topic of whether and how macrophages interact with nerves in the tongue needs further investigation.

It is also possible that tongue macrophages contribute directly or indirectly to instances of taste dysfunction. Various infections including Covid-19 or middle ear infections (*Mahmoud et al., 2021*; *Gautier and Ravussin, 2020*; *Michael and Raut, 2007*), different medical treatment regimens (*Wickham et al., 1999*) and aging (*Ogawa et al., 2017*) have all been associated with taste disruption. Taste bud maintenance relies on a continuous renewal of differentiated taste receptor cells (*Roper and Chaudhari, 2017*) and it has been shown that systemic inflammation like peripheral LPS injection increases TNFα and IL10 production by taste cells (*Feng et al., 2012*; *Feng et al., 2014*), inhibits taste progenitor cell proliferation and interferes with taste cell renewal (*Cohn et al., 2010*). Such a mechanism might explain taste disorders associated with infections in general. We showed that systemic inflammation also causes a direct response from tongue-resident macrophages. We did not investigate the precise contribution of inflammation to taste sensation, yet consider the possibility that tongue macrophages may release cytokines and potentially neurotoxic products that could cause nerve damage. The reverse has also been shown, that tissue macrophages can also mediate neuronal protection and therefore limit neuronal damage upon infection (*Matheis et al., 2020*). Whether and how the various tongue mononuclear phagocytes are involved in taste perception remains to be explored and could be of clinical relevance in conditions such as anorexia or cancer, where appetite- and weight-loss is often aggravated by taste-dysfunctions.

The question remains of the role of tongue immunity in cases where tongue homeostasis is disrupted. It was recently shown that tongue CD163+ macrophages infiltrate tumor tissue in squamous cell carcinoma at a high frequency (*Agarbati et al., 2021*). CD163+ macrophages were interpreted to be 'M2 macrophages', and increased infiltration correlated with worse outcomes compared to patients with a high infiltration of CD11c+ 'M1' macrophages (*Agarbati et al., 2021*). Regardless of whether the 'M1/M2' classification really applies to in vivo situations (*Nahrendorf and Swirski, 2016*), our data corroborate the notion of the existence of distinct subsets of tongue macrophages, which might respond differently to tumor-specific environmental cues.

In summary, we present a comprehensive catalog of immune cells in the murine tongue in physiological conditions and upon LPS-induced systemic inflammation. We identified two novel macrophage subsets in the tongue, namely tCX3CR1-MF and tFOLR2-MF, and place these findings in the context of mammalian macrophage biology. We hope that these data will encourage and support further investigations of the tongue as barrier and as an underrated immunological organ.

## Materials and methods
### Ethics statement, experimental reporting, and study design
This study was performed in strict accordance with national and international guidelines for the care and use of laboratory animals (Tierschutzgesetz der Bundesrepublik Deutschland, European directive

2010/63/EU, as well as GV-SOLAS and FELASA guidelines and recommendations for laboratory animal welfare). The animal experiment permission was obtained from the Landesamt für Gesundheit und Soziales (LAGeSo, Berlin). Most of the mice that were used in this study were also part of other experiments, in order to reduce animal experiments and suffering.

## Mice

We used the following mouse strains: C57BL/6 N wildytpe mice, B6(Cg)-Irf8tm1.2Hm/J ($Irf8^{-/-}$; Jackson laboratory, stock number: 018298), B6.SJL-Ptprca Pepcb/BoyJ (CD45.1/1; Jackson laboratory, stock number: 002014) and B6.129P2(Cg)-Cx3cr1tm1Litt/J ($Cx3cr1^{Gfp}$; *Jung et al., 2000*). For BM chimeras, 8–12 weeks old recipient animals (CD45.2/2) were lethally irradiated (950 rad) and reconstituted with $10^6$ BM cells isolated from CD45.1 expressing mice. The animals received Ciproxin in their drinking water for 10 days after irradiation. BM chimeras were analyzed 5 and 10 weeks after transfer. Mice were bred and housed at specific pathogen-free (SPF) animal facilities of the Max-Delbrück-Center for Molecular Medicine in Berlin, Germany or at the Weizmann Institute of Science in Rehovot, Israel. Mice were kept in standard conditions (22°C ± 1°C) under a 12 hr light cycle with experiments carried out during the 'lights-on' phase. Mice had access to a chow diet ad libitum and cages were lined with chip bedding and enriched with a mouse tunnel/igloo.

## Cell suspensions for fluorescence-activated cell sorting

Adult mice were anesthetized by intraperitoneal injection of 150 mg/kg body weight pentobarbital sodium (WDT) and intracardially perfused with PBS.

### Brain

Brains were dissected, and the olfactory bulb and cerebellum were removed. The remaining brain was minced and filtered through a 70-μm cell strainer in ice-cold high-glucose DMEM (Sigma). The cell suspension was centrifuged at 1200 rpm at 4 °C for 6 min and the pellet was resuspended in 5 ml of 40% Percoll (GE Healthcare) in PBS and transferred to a 15-ml tube. Tubes were centrifuged at 2000 rpm for 25 min at 14 °C with no acceleration nor break. Pellet was resuspended in 10 ml sorting buffer and centrifuged for 6 min at 1200 rpm and 4 °C. Leukocyte-containing cell pellets were further processed.

### Colon

Cells from $Cx3cr1^{Gfp/+}$ mice were isolated as previously described (*Aychek et al., 2015*). In brief, the intestine was removed and feces were flushed with cold PBS without calcium and magnesium. The intestine was longitudinally opened and cut into 0.5 cm pieces. Intestinal epithelial cells were removed by incubation with the HBSS containing 1 mM DTT, 2 mM EDTA, and 5% of fetal calf serum. The cell suspension was incubated on a shaker at 37 °C, 125 rpm for 40 min. The cell suspension was shortly vortexed and passed through a 100-μm mesh. The intestinal tissue pieces were transferred to a 50-ml Falcon tube containing 5 ml PBS with 5% FBS, 1 mg/ml of Collagenase VIII (Sigma) and 0.1 mg/ml DNase I. Samples were incubated on a shaker at 37 °C, 250 rpm for 40 min. Digested colon tissue was vortexed for 40 s, passed through a 70-μm mesh and cells were collected by centrifugation at 1200rpm for 10min at 4°C. Cells were stained after CD16/32 block with antibodies against CD45, CD11b, lineage (Ly6C, Ly6G, B220), and MHCII.

### Heart

The heart of $Cx3cr1^{Gfp/+}$ mouse was removed, the organ was chopped into small pieces and digested for 30 min at 37 °C in RPMI medium without fetal calf serum supplemented with 1 mg /ml Collagenase IV and 1 mg/ml DNase I. The digestion was stopped by addition of staining buffer (2 mM EDTA, 1% FCS in PBS) and the cell suspension was minced through a 100-μm cell strainer. After centrifugation at 1200 rpm, 4 °C for 6 min, supernatant was discarded and cells were blocked with anti-CD16/32 antibodies for 10 min, before anti-F4/80-biotin antibodies were added for 25 min on ice. After washing, the cell pellet was incubated with anti-biotin beads (Miltenyi) for 15 min on ice. Cells were washed again (1200 rpm, 4 °C for 6 min) and MACS was performed with LS columns (Miltenyi). After elution and washing of cells, lung cells were stained for CD45, streptavidin, dump (Ly6C, Ly6G), CD11b, MerTK, CD64, and MHCII.

## Lung

The lung was removed and cut into small pieces. Tissue was collected in RPMI medium without fetal calf serum supplemented with 1 mg/ml Collagenase A and 1 mg/ml DNase I. Tissue was digested for 30 min at 37 °C. The suspension was minced and filtered through a 100µm cell strainer. The cell suspension was centrifuged at 1200 rpm, 4 °C for 6 min. The supernatant was removed and cells were blocked with anti-CD16/32 antibodies for 10 min, before anti-CD45-biotin antibodies were added for 25 min on ice. Afterwards, cells were washed with staining buffer (2 mM EDTA, 1% FCS in PBS) at 1200 rpm, 4 °C for 6 min, supernatant was discarded and anti-biotin beads (Miltenyi) were added for 15 min. Cells were washed again (1200 rpm, 4 °C for 6 min) and MACS was performed with LS columns (Miltenyi). After elution and washing of cells, lung cells were stained for lineage (Ly6C, Ly6G), streptavidin, CD11b, CD64, CD11c, and SiglecF.

## Spleen

The spleens were removed from Bl6 mice and minced through a 100-µm cell strainer. After washing with staining buffer (2 mM EDTA, 1% FCS in PBS) at 1200 rpm, 4 °C for 6 min, supernatant was discarded and cells were blocked with anti-CD16/32 antibodies for 10 min, before anti-F4/80-biotin antibodies were added for 25 min on ice. After washing, the cell pellet was incubated with anti-biotin beads (Miltenyi) for 15 min on ice. Cells were washed again (1,200 rpm, 4 °C for 6 min) and MACS was performed with LS columns (Miltenyi). After elution and washing of cells, lung cells were stained for CD45, streptavidin, lineage (Ly6C, Ly6G, B220), CD11b, CD11c, and MHCII.

## Skin Langerhans cells

Ears were dissected and placed over 0.05% Trypsin with EDTA in PBS for 1.5–2 hr at 37 °C, until the epidermis could be peeled off using forceps. The epidermis was minced and the crude tissue suspension in sorting buffer was passed through a 100-µm cell strainer and centrifuged for 6 min at 1200 rpm and 4 °C. Cells were stained after CD16/32 block with antibodies against CD45, Gr1, CD11c and EpCam.

## Tongue leukocytes

Tongues were extracted, minced in 500 µL of PBS with 0.2 mg of DNAse I (Roche), 2.4 mg Collagenase IV (Gibco) and 0.15 mg (60 U) Hyaluronidase I (Sigma) and incubated for 45 min at 37 °C. Ten ml sorting buffer were added to the crude tissue suspension as it was passed through a 100-µm cell strainer and centrifuged for 6 min at 1200 rpm and 14 °C. Pellet was resuspended in 3 ml PBS, layered over 3 ml of Ficoll-Paque and centrifuged for 17 min at 2000 rpm and 14 °C with no acceleration nor break. The interface containing leukocytes was collected, 5 ml sorting buffer were added centrifuged for 6 min at 1200 rpm and 4 °C. Leukocyte-containing cell pellets were further processed. All tongue preparations were performed according to this protocol, if not otherwise stated.

## Tongue Langerhans cells

To separate the epithelium from the rest of the tongue and to dislodge Langerhans cells, extracted tongues were injected with 5 U/ml of Dispase II (Sigma) in HEPES buffered saline until completely distended. They were then incubated for 15 min at 37 °C and the epithelium layer was peeled off using forceps. The epithelium was minced in 500 µL of PBS with 0.2 mg of DNAse I (Roche), 4.8 mg Collagenase IV (Gibco) and 0.15 mg (60 U) Hyaluronidase I (Sigma) and incubated for 15 min at 37 °C. Ten ml sorting buffer were added to the crude tissue suspension as it was passed through a 100-µm cell strainer and centrifuged for 6 min at 1200 rpm and 4 °C. The pellet (enriched for tongue Langerhans cells) was then further processed accordingly.

## Flow cytometry and cell sorting

Cell suspensions were kept on ice. They were blocked with anti-CD16/32 (2.4G2) antibodies for 10 min and then stained for 20–25 min with antibodies against mouse CD45 (30-F11), CD45.2 (104), CD45.1 (A20), CD11b (M1/70), CD11c (N418), CD64 (X54-5/7.1), CX3CR1 (SA011F11), F4/80 (BM8), FOLR2 (10/FR2), EpCam (G8.8), Gr1 (RB6-8C5), IA/IE (M5/114.15.2), B220 (Ra3-6B2), Ly6C (HK1.4), Ly6G (1A8), LYVE1 (ALY7), MerTK (2B10C42), Siglec-F (E50-2440) and TIMD4 (RMT4-54). Antibodies were purchased from BioLegend or eBioscience. The gating strategy for all macrophage populations

is presented in *Figure 2—figure supplement 2*. Samples were washed in 2 ml sorting buffer and centrifuged for 6 min at 1200 rpm and 4 °C. Pellet was resuspended in sorting buffer and flow sorted using AriaI, AriaII or AriaIII (BD Biosciences, BD Diva Software) cell sorters. Flow cytometry analysis was performed on Fortessa or LSRII (BD Biosciences, BD Diva Software) and analyzed with FlowJo software v.10.7.1 (BD).

## scRNA-seq

### Experiment 1 (*Figure 1*)

Tongues were prepared as described in the Materials and methods section. Eight adult, female Bl6 mice were used for the isolation of interstitial cells with Collagenase IV / Hyaluronidase / DNase I digestion. Four additional adult, female Bl6 mice were used to isolate tongue Langerhans cells with Dispase digestion. Both preparations were pooled and CD45+ cells were FACS sorted and analyzed with the Chromium Single Cell 3' Reagent Kits v3.1.

### Experiment 2 (*Figure 1—figure supplement 1a*)

scRNA-seq was performed on CD45+ tongue hematopoietic cells isolated from 10 Bl6 female mice. Tongues were extracted, minced in 500 μL of PBS and digested with 0.2 mg of DNAse I (Roche) and 1 mg/ml Collagenase IV (Gibco) and no enrichment for Langerhans cells was performed. 10 ml sorting buffer were added and tissue suspension was passed through a 100-μm cell strainer and centrifuged for 6 min at 1,200 rpm and 14 °C. FACS-purified CD45+ cells were used for scRNA-seq. scRNA-seq was performed with the Chromium Single Cell 3' Reagent Kits v2.

### Experiment 3 (*Figure 4*)

Cell isolation of LPS-injected female Bl6 mice was performed as mentioned for experiment 1. Six mice were used for interstitial cell isolation (Collagenase IV / Hyaluronidase / DNase I digestion) and 4 mice for Langerhans cell extraction (Dispase digestion). Both samples were pooled and CD45+DAPI- cells were analyzed with Chromium Single Cell 3' Reagent Kits v3.1.

### Experiment 4 (*Figure 5*)

Seven mice were used for interstitial cell isolation by Collagenase IV / Hyaluronidase / DNase I digestion and no enrichment for Langerhans cells was performed. CD45+DAPI- FACS-sorted cells were analyzed with Chromium Single Cell 3' Reagent Kits v3.1.

### Experiment 5 (*Figure 6—figure supplement 3*)

Cell isolation of Irf8-/- female Bl6 mice was performed as mentioned above (Collagenase IV / Hyaluronidase / DNase I). Five mice were used for interstitial cell isolation. No additional Langerhans cell extraction was performed for this experiment. CD45+DAPI- cells were analyzed with Chromium Single Cell 3' Reagent Kits v3.1.

## RNA isolation and cDNA synthesis for bulk RNA-Seq

500–20,000 sorted cells were lysed with 100 μl of lysis/binding buffer (Life Technologies), snap-frozen on dry ice and stored at –80 °C until further use. mRNA purification was performed with the Dynabeads mRNA DIRECT Purification Kit (Life Technologies) according to the manufacturer's guidelines. MARS-seq barcoded RT primers were used for reverse transcription with the Affinity Script cDNA Synthesis Kit (Agilent) in a 10 μl reaction volume.

### Bulk RNA-sequencing

The MARS-seq protocol was used for bulk RNA sequencing (*Jaitin et al., 2014*). After reverse transcription, samples were analyzed by qPCR and samples with similar Ct values were pooled. Samples were treated with Exonuclease I (New England BioLabs (NEB)) for 30 min at 37 °C and for 10 min at 80 °C followed by a 1.2 X AMPure XP beads (Beckman Coulter) cleanup. The second strand synthesis kit (NEB) at 16 °C for 2 hours was used for cDNA synthesis followed by a 1.4 X AMPure XP bead

cleanup. In vitro transcription (IVT) was performed at 37 °C for 13–16 h with the HiScribe T7 RNA Polymerase kit (NEB). The remaining DNA was digested by Turbo DNase I (Life Technologies) treatment at 37 °C for 15 min followed by a 1.2 X AMPure XP bead cleanup. RNA fragmentation (Invitrogen) was performed at 70 °C and the reaction was stopped after 3 min with Stop buffer (Invitrogen) followed by a 2 X AMPure XP bead cleanup. Ligation of the fragmented RNA to the MARS-seq adapter was performed at 22 °C for 2 h with T4 RNA ligase (NEB) followed by a 1.5 X AMPure XP bead cleanup. A second reverse transcription reaction was performed with MARS-seq RT2 primer and the Affinity Script cDNA Synthesis Kit (Agilent Technologies) followed by 1.5 X AMPure XP bead cleanup. Finally, the library was amplified using P5_Rd1 and P7_Rd2 primers and the Kapa HiFi Hotstart ready mix (Kapa Biosystems) followed by a 0.7 X AMPure XP bead cleanup. Fragment size was measured using a TapeStation (Agilent Technologies) and library concentrations were measured with a Qubit fluorometer (Life Technologies). The samples were sequenced using a NextSeq 500 system (Illumina).

## Microscopy

### Tissue preparation

Mice were deeply anesthetized with a combination of 150 mg/kg body weight pentobarbital sodium (WDT) and perfused transcardially for 3 min with ice-cold 0.9% NaCl solution and for 5 min with 4% paraformaldehyde (PFA) in 0.1 M phosphate buffer (PB). Tongues were post-fixed overnight in 4% PFA in 0.1 M PB. They were then cryoprotected by 3 x overnight incubations in 30 % w/v sucrose in 0.1 M PB, cryosectioned with a cryostat (35 µm sagittal) and mounted directly to slides for staining.

### Immunohistochemistry

35 µm sagittal tongue sections from the tongue midline were washed for 5 min at RT in TBS (42 mM Tris HCl, 8 mM Tris Base, 154 mM NaCl; pH 7.4). Sections were blocked in 20% normal donkey serum (NDS) in TBS-T for 1 hr at RT and incubated overnight at 4 °C in primary antibody diluted in TBS-T +2% NDS/NGS. Primary antibodies against CD31 (Millipore MAB13982; 1:200), CD68 (Biolegend 137002; 1:100), GFP (AbCam; ab13970; 1:200), LYVE1 (ReliaTech 103-PA50AG; 1:500), Podoplanin (Biolegend 156202; 1:50), TUJ1 (AbCam ab18207; 1:200) were used. Sections were then washed 3 × 30 min in TBS-T and incubated overnight at 4 °C with appropriate secondary antibodies raised in goat and conjugated to Alexa Fluor dyes (Invitrogen) were diluted 1:500 or 1:1,000 in TBS-T +2% NDS/NGS. Sections were washed 3 × 30 min in TBS-T and nuclei were stained with 0.5 mg/ml DAPI in TBS. After washing, sections were mounted with FluorSave reagent (Calbiochem).

### Imaging and quantification

Imaging was performed using a Leica SP5 TCE. Four regions (R1-R4; *Figure 3—figure supplement 1*) from three consecutive sagittal 35 µm slices at the tongue midline were imaged at 775.76 × 734.76 µm dimensions and over a depth of 30 µm with 1.5 µm z-stack intervals. All image processing was done with Fiji (*Schindelin et al., 2012*) and quantifications were performed with Imaris Microscopy Image Analysis Software (Bitplane). Regions of interest (ROI: muscle or lamina propria) were manually drawn onto each z-stack image and the software's built-in surface module extrapolated the volume of the respective ROI. Then, the built-in surface detection algorithm was used to identify cells (LYVE1+ or *Cx3cr1*-GFP+) in each ROI, so that we could calculate number of cells and mean cell volumes per ROI volume.

### Lipopolysaccharide induced systemic inflammation

Mice were injected intraperitoneally with 1 mg/kg LPS (*E. coli* 0111:B4) in 200 µl PBS 6 hr prior to sacrifice.

### Cell proliferation assay (EdU)

The EdU Click 488 Kit from BaseClick was used. In brief, 0.5 mg/g of EDU in PBS was injected intraperitoneally to adult mice or subcutaneously to pups 15 hr before to sacrifice. Single cell suspensions were obtained from tongues (see relevant section) and cells were processed according to the manufacturer's instructions.

## Sequencing data analysis

### Bulk sequencing data analysis

Using the fastq files, the reads were deduplicated based on their UMIs. Subsequently, STAR (version 2.5.3 a) was applied for the alignment of reads to the mouse genome (mm9). Quantification of reads by htseq-count (version 1.0) yielded the input expression matrix for DESeq2 (version 1.26.0), used to identify differentially expressed genes for each group. More specifically, each group was compared against all other groups to obtain DE marker genes.

## Single-cell sequencing data analysis

### Preprocessing and integration

The data was sequenced with 10 x Genomics (version v3.1). For alignment to mm10 and quantification Cell Ranger Single Cell Software was used. For additional preprocessing and downstream analysis Seurat (v4) was applied. First, based on the UMI counts, cells were filtered based on the number of detected genes and the proportion of mitochondrial gene counts. All cells with more than 10% mitochondrial gene count were removed. For the number of detected genes, a sample specific filtering value was applied, for experiment 2 (10x Chromium v2) the accepted range was between 300 and 2000, for the remaining samples (10 x Chromium v3) a minimum of 500 was required while the maximum cutoff ranged from 5000 (experiment 1) over 5500 (experiment 3)–6000 (experiment 4). The data was normalized and the 4000 most valiable genes were selected. Subsequently, the data was analyzed together by applying Seurats integration function (FindIntegrationAnchors) and making use of the pre-computed anchors (FindIntegrationAnchors), that is genes used to map the cells from different samples. Analogously, experiment 5 (*Irf8*-deficient cells) and experiment 1 were integrated. For the IRF8 sample, the range of detected genes for included cells was set to 500–5500 and cells with more than 10% mitochondrial gene count were removed.

### Dimension Reduction, unsupervised clustering and marker detection

A principal component analysis provided the basis for the computation of a UMAP and an unsupervised clustering of the cells. The resolution parameter for FindClusters was set to 0.42 (0.25 for experiment 5). Conserved cluster markers for all samples were computed using FindConservedMarkers with the detection method set to 'MAST'. Genes differentially expressed between clusters of different samples were detected analogously using FindClusters.

### Cell cluster annotation, trajectory inference, and signature enrichment

To assign cell types to the clusters identified, we applied SingleR (version 1.6.1; *Aran et al., 2019*), using the ImmGen database as annotation resource. Slingshot (version 2.0.0; *Street et al., 2018*) was applied for trajectory inference. To this end, we provided slingshot with the newborn-specific cluster as starting point. To shed more light on the macrophagic versus dendritic nature of the cells in the central clusters, we computed the enrichment of macrophage-specific as well as dendritic gene sets by use of GSVA (version 1.40.1; *Hänzelmann et al., 2013*).

## Acknowledgements

We would like to thank Victoria Malchin, Jermaine Voß and Sarah Jaksch for excellent technical support, as well as the MDC animal facility, especially Juliette Bergemann, the MDC FACS core unit, particular Dr. Hans-Peter Rahn, and the MDC genomic core facility, especially Caroline Braeuning. We thank Dr. Anca Margineanu, Dr. Sandra Cristina Carneiro Raimundo and the Advanced Light Microscopy Technology Platform of the MDC for the general and technical support. C.S.N.K. was supported by grants from the German Research Foundation (FOR2599 project 5 - KL 2963/5-2; KL 2963/2-1 and KL 2963/3-1) and the European Research Council Starting Grant (ERCEA; 803087). A.M received a Heisenberg fellowship from the DFG (MI1328/3-1).

# Additional information

## Competing interests

Simon Yona: Reviewing editor, *eLife*. The other authors declare that no competing interests exist.

## Funding

| Funder | Grant reference number | Author |
|---|---|---|
| Deutsche Forschungsgemeinschaft | MI1328 | Alexander Mildner |
| Deutsche Forschungsgemeinschaft | KL 2963/5-2 | Christoph SN Klose |
| European Research Council | ERCEA 803087 | Christoph SN Klose |
| Deutsche Forschungsgemeinschaft | KL 2963/2-1 | Christoph SN Klose |
| Deutsche Forschungsgemeinschaft | KL 2963/3-1 | Christoph SN Klose |

The funders had no role in study design, data collection and interpretation, or the decision to submit the work for publication.

## Author contributions

Ekaterini Maria Lyras, Conceptualization, Data curation, Formal analysis, Investigation, Methodology, Writing – original draft, Writing – review and editing; Karin Zimmermann, Data curation, Methodology, Software, Visualization; Lisa Katharina Wagner, Data curation, Investigation; Dorothea Dörr, Data curation, Investigation, Methodology, Writing – review and editing; Christoph SN Klose, Formal analysis; Cornelius Fischer, Data curation, Formal analysis; Steffen Jung, Simon Yona, Avi-Hai Hovav, Achim Leutz, Resources, Writing – review and editing; Werner Stenzel, Steffen Dommerich, Methodology, Resources; Thomas Conrad, Resources, Software; Alexander Mildner, Conceptualization, Data curation, Formal analysis, Funding acquisition, Investigation, Methodology, Project administration, Resources, Supervision, Visualization, Writing – original draft, Writing – review and editing

## Author ORCIDs

Ekaterini Maria Lyras ⓘ http://orcid.org/0000-0003-3269-0854
Steffen Jung ⓘ http://orcid.org/0000-0003-4290-5716
Simon Yona ⓘ http://orcid.org/0000-0002-3984-2008
Alexander Mildner ⓘ http://orcid.org/0000-0002-2019-8427

## Ethics

This study was performed in strict accordance with national and international guidelines for the care and use of laboratory animals (Tierschutzgesetz der Bundesrepublik Deutschland, European directive 2010/63/EU, as well as GV-SOLAS and FELASA guidelines and recommendations for laboratory animal welfare). The animal experiment permission was obtained from the Landesamt für Gesundheit und Soziales (LAGeSo, Berlin).

## Decision letter and Author response

Decision letter https://doi.org/10.7554/eLife.77490.sa1
Author response https://doi.org/10.7554/eLife.77490.sa2

---

# Additional files

## Supplementary files

• Transparent reporting form

## Data availability

Data that were generated within this study are deposited in Gene Expression Omnibus (GEO) with the accession code GSE205162.

The following dataset was generated:

| Author(s) | Year | Dataset title | Dataset URL | Database and Identifier |
|---|---|---|---|---|
| Lyras E, Zimmermann K, Wagner L, Dörr D, Fischer C, Jung S, Leutz A, Mildner A | 2022 | Tongue immune compartment analysis reveals spatial macrophage heterogeneity | https://www.ncbi.nlm.nih.gov/geo/query/acc.cgi?acc=GSE205162 | NCBI Gene Expression Omnibus, GSE205162 |

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
