## [Editor Report]

Here, the authors map the cellular landscape in the tongue, an understudied immunological organ, with a main focus on tissue-resident myeloid cells under homeostatic and inflammatory conditions. They identify two major subsets of macrophages, which occupy distinct anatomical niches and develop from local precursors, while under immune compromised conditions they can also be replenished from circulating hematopoietic precursors. These findings provide an important basis for future investigations of the tongue immune function in the context of infection, inflammation, and neoplastic diseases.

---

## [Decision Letter]

**Decision letter after peer review:**

Thank you for submitting your article "Tongue immune compartment analysis reveals spatial macrophage heterogeneity" for consideration by *eLife*. Your article has been reviewed by 2 peer reviewers, and the evaluation has been overseen by a Reviewing Editor and Carla Rothlin as the Senior Editor. The reviewers have opted to remain anonymous.

Essential revisions:

1) The systemic LPS challenge should be better justified and explain why choosing an ip model of LPS administration with minor impact on studied populations. Flow cytometry of blood and tongue will strengthen the study as well as expand on the putative differential role of the two macrophage subsets in antimicrobial defence.

2) Provide fluorescence microscopy images around P10 will complement the study to follow the transition of CX3CR1+ cells.

*Reviewer #2 (Recommendations for the authors):*

I have only a few recommendations:

Figure 2 (in connection with Figure 5): Cells of the FOLR2+ macrophage subset express various levels of CX3CR1 (monitored with the GFP reporter). As shown in Figure 5, the levels of CX3CR1 decrease over time. Do CX3CR1(hi) and CX3CR1(lo) FOLR2+ macrophages differ in expression of any of the markers tested by FACS and/or transcriptomics? Can any additional information be gained about the differentiation process of the cellular subset?

Figure 4: The authors assessed the response of the two macrophage subsets to systemic LPS challenge. Given the constant exposure of the tongue to commensal and pathogenic microbes (and viruses) in the oral cavity, it seems of higher importance to probe the reactivity of the studied macrophage subsets to oral microbes. While such experiments might go beyond the scope of the current study, the authors should at least consider expanding their discussion on the putative differential role of the two macrophage subsets in antimicrobial defence.

Figure 5c: fluorescence microscopy images are shown for E14, p0 and p3 (not for p10 as indicated in the text). Showing data for later time points (around p10 when CX3CR1+FOLR2+ cells start to lose CX3CR1 expression) would be of interest to follow how the transition of CX3CR1+ cells becoming restricted to the LP.

Figure 6e: The authors conclude from the developmental trajectory analysis that both terminally differentiated macrophage subsets (cluster 0 and 5) derive from a common precursor. However, the CX3CR1+ FOLR2- subset appears as early as E17.5 as a separate subset. Do the CX3CR1+ FOLR2- cells differ by any means between E17.5, p3 and adulthood (e.g. parameters analyzed by flow cytometry or genes analyzed by RNAseq)? Which factors drive the developmental process? Is the microbiota involved?

Among the lymphoid populations, the authors detected Tregs, ILC2 and Rora+ T cells. Did they also find other T cells? Of special interest are Th17 cells, which are found in high numbers in the gingiva of normal mice (doi: 10.1016/j.immuni.2016.12.010). To what extent do these to oral tissue compartments differ or compare, at least with respect to selected cell populations?

---

## [Author Response]

Reviewer #2 (Recommendations for the authors):I have only a few recommendations:Figure 2 (in connection with Figure 5): Cells of the FOLR2+ macrophage subset express various levels of CX3CR1 (monitored with the GFP reporter). As shown in Figure 5, the levels of CX3CR1 decrease over time. Do CX3CR1(hi) and CX3CR1(lo) FOLR2+ macrophages differ in expression of any of the markers tested by FACS and/or transcriptomics? Can any additional information be gained about the differentiation process of the cellular subset?

To answer the question about the maturation of tFOLR2+ macrophages, we performed a new latent time analysis in which we analyzed potential developmental gene expression programs and the possible underlying molecular mechanism (new Figure 6—figure supplement 2a+b). We identified a general down-regulation of MHC-related genes from immature to mature tFOLR2+ macrophages and an increased gene expression that show STAT3 binding motifs. A similar analysis was performed for the tCX3CR1-MF population, but we could not detect a developmental program in these cells (new Figure 6—figure supplement 2c+d).

Figure 4: The authors assessed the response of the two macrophage subsets to systemic LPS challenge. Given the constant exposure of the tongue to commensal and pathogenic microbes (and viruses) in the oral cavity, it seems of higher importance to probe the reactivity of the studied macrophage subsets to oral microbes. While such experiments might go beyond the scope of the current study, the authors should at least consider expanding their discussion on the putative differential role of the two macrophage subsets in antimicrobial defence.

We agree with the Reviewers’ criticism which is in line with Reviewer #1’s first point and we hope that we have sufficiently clarified our rationale for systemic LPS administration in our response above. We also agree that investigating the role of oral microbiota on the development of tongue immune cells would indeed be very interesting, especially since the maturation of these cells seems to coincide with lactation and weaning. Indeed, Caputcha et al*.,* 2018, revealed that local microbiota has a major impact on the development of mucosal LCs and this could also be the case for myeloid cells of the tongue. It would be interesting to investigate tongue immune cell development in germ-free mice, for example, but we do not currently have permission for such experiments as part of this study. We have added relevant discussion in the main text of the manuscript.

Figure 5c: fluorescence microscopy images are shown for E14, p0 and p3 (not for p10 as indicated in the text). Showing data for later time points (around p10 when CX3CR1+FOLR2+ cells start to lose CX3CR1 expression) would be of interest to follow how the transition of CX3CR1+ cells becoming restricted to the LP.

We added microscopy of p11 tongues to Figure 5c, as requested. At this time point, *Cx3cr1*-GFP+ cells were evident in the lamina propria, but GFP+ were still clearly detectable in deeper muscular tissues. The strict arrangement of *Cx3cr1*-GFP^+^Lyve1^-^ cells in the lamina propria is therefore a feature of the adult, but not the developing tongue. We added a respective description of the results to the new version.

Figure 6e: The authors conclude from the developmental trajectory analysis that both terminally differentiated macrophage subsets (cluster 0 and 5) derive from a common precursor. However, the CX3CR1+ FOLR2- subset appears as early as E17.5 as a separate subset. Do the CX3CR1+ FOLR2- cells differ by any means between E17.5, p3 and adulthood (e.g. parameters analyzed by flow cytometry or genes analyzed by RNAseq)? Which factors drive the developmental process? Is the microbiota involved?

At E17.5 prior to birth and at birth (p0 data not shown), there are no CX3CR1-GFPFolr2^+^ cells in the murine tongue. This population starts to be present at p3 (Figure 5a). On the contrary and as mentioned correctly by the Reviewer, the *Cx3cr1*-GFP^+^Folr2^-^ population is already present at E17.5 and is maintained through adulthood. As suggested by the Reviewer, we now compared the expression profile of p3 and adult tCX3CR1-MF to each other and now present this in Figure 6—figure supplement 1. We also discuss the microbiota as a possible factor that might influence the development of tongue macrophages.

Among the lymphoid populations, the authors detected Tregs, ILC2 and Rora+ T cells. Did they also find other T cells? Of special interest are Th17 cells, which are found in high numbers in the gingiva of normal mice (doi: 10.1016/j.immuni.2016.12.010). To what extent do these to oral tissue compartments differ or compare, at least with respect to selected cell populations?

To answer this question, we filtered cells that have a T lymphocyte but lack macrophage or DC signature and performed a new UMAP analysis. With this reclustering, T cells could further be separated into 11 distinct subsets in which we could identify ILCs, γδ T cells, central T cells but not Th17 cells. The new analysis of T cells is shown in new Figure 1b.